# Differential regulation of BIRC2 and BIRC3 expression by inflammatory cytokines and glucocorticoids in pulmonary epithelial cells

**Andrew Thorne[1], Akanksha Bansal[1], Amandah Necker-Brown[1], Mahmoud M. Mostafa[1], Alex Gao[1], Andrei Georgescu[1], Cora Kooi[2], Richard Leigh[1,2], Robert Newton⦿[1]***

**1** Department of Physiology & Pharmacology, Snyder Institute for Chronic Diseases, Lung Health Research Group, Cumming School of Medicine, University of Calgary, Calgary, Alberta, Canada, **2** Department of Medicine, Snyder Institute for Chronic Diseases, Lung Health Research Group, Cumming School of Medicine, University of Calgary, Calgary, Alberta, Canada

* rnewton@ucalgary.ca

**Data Availability Statement:** All relevant data are within the paper and its Supporting Information files.

## Abstract

Roles for the baculoviral inhibitor of apoptosis repeat-containing (BIRC) genes, BIRC2 and BIRC3, may include signaling to the inflammatory transcription factor, nuclear factor-κB (NF-κB) and protection from cell death. However, distinct functions for each BIRC are not well-delineated. Given roles for the epithelium in barrier function and host defence, BIRC2 and BIRC3 expression was characterized in pulmonary epithelial cell lines and primary human bronchial epithelial cells (pHBECs) grown as undifferentiated cells in submersion culture (SC) or as highly differentiated cells at air-liquid interface (ALI). In A549 cells, inter-leukin-1β (IL1B) and tumor necrosis factor α (TNF) induced BIRC3 mRNA (~20-50-fold), with maximal protein expression from 6–24 h. Similar effects occurred in BEAS-2B and Calu-3 cells, as well as SC and ALI pHBECs. BIRC2 protein was readily detected in unstimulated cells, but was not markedly modulated by IL1B or TNF. Glucocorticoids (dexamethasone, budesonide) modestly increased BIRC3 mRNA and protein, but showed little effect on BIRC2 expression. In A549 cells, BIRC3 mRNA induced by IL1B was unchanged by glucocorticoids and showed *supra*-additivity with TNF-plus-glucocorticoid. *Supra*-additivity was also evident for IL1B-plus-budesonide induced-BIRC3 in SC and ALI pHBECs. Using A549 cells, IL1B- and TNF-induced BIRC3 expression, and to a lesser extent, BIRC2, was prevented by NF-κB inhibition. Glucocorticoid-induced BIRC3 expression was prevented by silencing and antagonism of the glucocorticoid receptor. Whereas TNF, but not IL1B, induced degradation of basal BIRC2 and BIRC3 protein, IL1B- and TNF-induced BIRC3 protein remained stable. Differential regulation by cytokines and glucocorticoids shows BIRC2 protein expression to be consistent with roles in rapid signaling events, whereas cytokine-induced BIRC3 may be more important in later effects. While TNF-induced degradation of both BIRCs may restrict their activity, cytokine-enhanced BIRC3 expression could prime for its function. Finally, shielding from glucocorticoid repression, or further enhancement by glucocorticoid, may indicate a key protective role for BIRC3.

**Funding:** This work was supported by: RN grants: Canadian Institutes of Health Research (CIHR) (MOP 125918, PJT 156310) and Natural Sciences and Engineering Research Council of Canada (NSERC) discovery grant (RGPIN- 2016-04549); AB studentships: Eyes High Doctoral Scholarship and Eleanor Mackie Doctoral Scholarship in Women's Health from University of Calgary; MMM studentships: NSERC Postgraduate Scholarship - Doctoral, Queen Elizabeth II Doctoral scholarship, and The Lung Association—Alberta & NWT studentship awards. The funders had no role in study design, data collection and analysis, decision to publish, or preparation of the manuscript.

**Competing interests:** The authors have declared that no competing interests exist.

## Introduction

The products of the baculoviral inhibitor of apoptosis (IAP) repeat-containing (BIRC) genes, *BIRC2* and *BIRC3*, are cell signaling regulators involved in innate immune responses and are often referred to as cellular IAP 1 and 2, respectively [1,2]. Believed to prevent apoptosis, BIRC2 and BIRC3 are members of a conserved IAP protein family, the defining feature of which is the presence of one-to-three N-terminal tandem baculovirus IAP repeats [3]. BIRC2 and BIRC3 also possess a carboxy-terminal really interesting new gene (RING) domain that confers E3 ubiquitin ligase activity. This may result in mono- or polyubiquitylation of target proteins leading to proteasome-mediated degradation or the formation of scaffolds that promote intracellular signaling [4–6].

Probably the most widely characterized roles for BIRC2 and/or BIRC3 are in the regulation of canonical and non-canonical signaling to the inflammatory transcription factor, nuclear factor-κB (NF-κB) [7–10]. Activation of signaling by the inflammatory cytokine, tumor necrosis factor-α (TNF) involves the TNF receptor 1 (TNFR1) complex, where BIRC2 and/or BIRC3 are suggested to serve as E3 ubiquitin ligases for receptor-interacting serine/threonine-protein kinase 1 (RIPK1) [11,12]. In this scenario, K63-linked ubiquitin scaffolds are attached to RIPK1 to enable canonical NF-κB signaling via the inhibitor of κB kinases (IKKs) [11]. Activated NF-κB not only induces expression of inflammatory cytokines, chemokines and inflammatory enzymes, but also numerous anti-apoptotic genes [13–15]. Many of these play pivotal roles in inhibiting caspase-mediated apoptosis and are essential for cell survival. For example, in the absence of BIRC2 or BIRC3, RIPK1 is thought to recruit to a secondary cytoplasmic signaling complex that activates pathways leading to cell death [6]. In contrast to promoting K63-linked ubiquitin chains leading to IKK activation, BIRC2 and BIRC3 are also suggested to maintain low levels of NF-κB-inducing kinase (NIK) in resting cells via the formation of K48-linked ubiquitin chains leading to proteasomal degradation of NIK [12,16]. Activation of TNF receptor superfamily members, or BIRC2 and BIRC3 inhibition, appears to stabilize NIK and thereby promotes non-canonical NF-κB signaling [9]. However, in addition to these roles. BIRC2 and/or BIRC3 are also implicated in signaling downstream of toll-like receptors [17], nucleotide-binding oligomerization domain-containing receptors [18], retinoic acid-inducible gene 1 [19], as well as in inflammasome activation [20]. Furthermore, BIRC2 and/or BIRC3 are suggested to directly regulate transcription factors such as IRF1 [21]. Thus, in addition to their apparently interchangeable, or redundant, roles, the above studies also highlight a lack of clarity as to the roles of these two BIRC proteins.

Pattern recognition receptors (PRRs) detect and respond to the wide array of pathogen-associated molecular patterns (PAMPs) and damage-associated molecular patterns (DAMPs) that are produced upon tissue damage, cell death or cellular stress [22,23]. In the airway epithelium, this promotes the release of cytokines, chemokines and antimicrobial peptides that can drive inflammation in diseases such as asthma [23,24]. As signaling from many PRRs converges on NF-κB, it is perhaps not surprising that polymorphisms in *BIRC3* have been associated with asthma [25]. In addition, BIRC3 expression correlates with markers of severe asthma [26], suggesting BIRC3 could contribute to a more prolonged, or more severe, inflammation. In contrast, BIRC3 may be protective in influenza virus infections and thus transcriptional induction of BIRC3 by inflammatory triggers may help protect from viral-induced cell death [27]. The airway epithelium is also a key target for inhaled therapies, including glucocorticoids, which represent the mainstay therapy in asthma management [28–30]. While glucocorticoids reduce the expression of many genes that are upregulated by NF-κB, we [31,32], and others [33–36] have consistently found BIRC3 to be a *bona fides* glucocorticoid-response gene in epithelial cells and in human lung tissue following corticosteroid administration [37].

Given the unclear, potentially redundant, roles of BIRC2 and BIRC3 in the regulation of inflammatory signaling and gene expression, pulmonary epithelial cell lines and primary human bronchial epithelial cells (pHBECs) were used to characterize the expression and regulation of both BIRC2 and BIRC3 in response to inflammatory cytokines and glucocorticoids. The data reveal largely constitutive expression for BIRC2 protein, whereas BIRC3 protein was highly induced by pro-inflammatory cytokines via a NF-κB-dependent mechanism. BIRC3 expression was also induced by glucocorticoids acting on GR and in combination with inflammatory stimuli, this resulted in maintained, often enhanced, expression relative to the inflammatory stimulus alone. As the stability of BIRC2 and BIRC3 proteins was profoundly reduced by TNF, but not IL1B, these data uncover discrete patterns of regulation and expression for each gene. Since this is likely to underpin differential functions for each protein, the data provide a framework for further investigation of BIRC2 and BIRC3 function.

## Materials and methods

### Stimuli, drugs and inhibitors

Recombinant human IL1B, TNF and IFNG (R&D Systems, Minneapolis, MN) were dissolved in PBS containing 0.1% bovine serum albumin (Sigma-Aldrich, St. Louis, MO). Budesonide (AstraZeneca, Mölndal, Sweden), dexamethasone (Steraloids, Newport, RI), MG-132 (Millipore Sigma, St. Louis, MO), MG-262 (APExBIO, Houston, TX), PR-171 (Adooq BioScience, Irvine, CA), E-64 (APExBIO, Houston, TX), PS-1145, ML-120B (both Sigma-Aldrich) and Org34517 (gift from Chiesi Farmaceutici, Parma, Italy) were dissolved in dimethyl sulfoxide as stocks of 10 mM. Cycloheximide (Sigma-Aldrich) was dissolved in water at 10 mg/ml.

### Cell lines, culture and reporter lines

Human pulmonary type II A549 cells (CCL-185, American Type Culture Collection (ATCC), Manassas, VA) were grown in Dulbecco's modified Eagle's medium (DMEM) supplemented with 10% fetal bovine serum (FBS) and 2 mM L-glutamine (all Thermo Fisher Scientific, Burlington, ON, Canada). Human bronchial epithelial, BEAS-2B, cells (CRL-9609; ATCC, Manassas, VA)) were grown in DMEM/F12 (Thermo Fisher Scientific) supplemented with 10% FBS and 2 mM L-glutamine. The Calu-3 epithelial cell line (HTB-55; ATCC) was grown in Eagle's Minimum Essential Medium (ATCC) supplemented with 10% FBS. The NF-κB-dependent reporter, 6κBtkluc.neo, contains three tandem repeats of the sequence 5′-AGC TTA CAA GGG ACT TTC CGC TGG GGA CTT TCC AGG GA-3′, which harbors two copies of the NF-κB binding site (underlined), was stably transfected into A549 cells as described [38]. A549 cells containing the 2×GRE-dependent reporter, which is based on the parent vector pGL3.neo.TATA, driven by two consensus GRE sites (underlined), 5′-GCT GTA CAG GAT GTT CTA GGC TGT ACA GGA TGT TCT AG-3′, were as previously described [39]. All cells were grown in submersion culture at 37°C in 5% CO2:95% air and passaged when 90–95% confluent. Prior to experiments, cells were incubated overnight with serum-free basal media to arrest growth.

### Culture of primary human bronchial epithelial cells in submersion culture and at air-liquid interface

Primary human bronchial epithelial cells (pHBEC) were isolated from non-transplantable normal human lungs that were obtained via a retrieval service at the International Institute for the Advancement of Medicine (Edison, NJ), as previously described [40]. In each case, no identifying donor information was provided and local ethics approval was obtained from the Conjoint

Health Research Ethics Board of the University of Calgary (Protocol # REB15-0336). pHBECs as submersion culture (pHBEC-SC) were grown in bronchial epithelial cell growth medium (BEGM) (Lonza, Basel, Switzerland) containing all SingleQuots supplements (Lonza, Basel, Switzerland), as described [40]. pHBECs as air-liquid interface cultures (pHBEC-ALI) were grown to confluence in PneumaCult-EX cell growth medium (05009; StemCell Technologies, Vancouver, BC, Canada). Growth medium was then removed from the apical surface to expose pHBECs to air and the basal growth medium replaced with differentiation medium (Pneuma-Cult-ALI basal medium (05002; StemCell Technologies). pHBEC-ALI's were washed apically with Dulbecco's phosphate-buffered saline (D-PBS) (Thermo Fisher Scientific) to remove excess mucus resulting from goblet cell differentiation. This results in pHBEC-ALI cultures that accurately recapitulate a pseudostratified epithelium composed of ciliated cells, goblet cells, and progenitor basal cells, see [41]. pHBEC-ALI were fed basally with PneumaCultALI basal medium, with no supplements, prior to experiments. All cells were incubated at 37˚C in 5% $CO_2$:95% air.

## Reverse Transcriptase PCR

Total RNA was extracted using the NucleoSpin RNA Extraction kit (MN-740955; D-Mark Biosciences, Macherey-Nagel, Bethlehem, PA) and 0.5 μg was used for cDNA synthesis (qScript cDNA Synthesis Kit, CA101414-098; Quanta Biosciences, Gaithersburg, MD). Resultant cDNA was diluted 1:5 with RNase-free water and quantitative PCR (qPCR) performed on 2.5 μl using Fast SYBR Green Master Mix (Applied Biosystems, Foster City, CA) with primer pairs specific for genes of interest. StepOnePlus (Applied Biosynthesis) or QuantaStudio3 (Thermo Fisher Scientific) instruments were utilized for qPCR and relative cDNA concentrations were calculated from standard curves generated by half-log serial dilution of a stimulated cDNA sample analyzed at the same time as experimental samples. Amplification conditions were: 95˚C for 20 s, then 40 cycles of 95˚C for 3 s and 60˚C for 30 s. Primers (5'- 3'): BIRC3 (Forward: `CCG TCA AGT TCA AGC CAG TTA CCC` and Reverse: `AGC CCA TTT CCA CGG CAG CA`), BIRC2 (Forward: `CTA GTC TGG GAT CCA CCT CTA A` and Reverse: `GTT CCA AGG TGG GAG ATA ATG`), and GAPDH (Forward: `TTC ACC ACC ATG GAG AAG GC` and Reverse: `AGG AGG CAT TGC TGA TGA TCT`) were designed using Primer BLAST (NCBI) and were synthesized by the DNA synthesis lab at the University of Calgary. Primer specificity was determined using dissociation (melt) curve analysis: 95˚C for 15 s, 60˚C for 20 s followed by ramping to 95˚C with fluorescence measurement every 2.5 degrees. A single peak in the change of fluorescence with temperature was taken to indicate primer specificity.

## Western blot analysis

Western blotting was carried out as previously described [42]. Following cell lysis, proteins were size fractionated by sodium dodecyl sulphate-polyacrylamide gel electrophoresis (SDS-PAGE). Proteins were then transferred to nitrocellulose membranes before blocking and incubation with primary antibodies against BIRC3 (cIAP2 #3130; Cell Signaling Technologies, Danvers, MA), BIRC2 (cIAP1 #7065; Cell Signaling Technologies), RELA (p65 sc-8008; Santa-Cruz, Dallas, TX), GR (PA1-511A; Thermo Fisher Scientific), K48-linkage specific polyubiquitin (#8081; Cell Signaling Technologies), GAPDH (MCA4739; Bio-Rad, Hercules, CA) overnight at 4˚C. Membranes were washed in tris-buffered saline with tween (TBS-T) followed by incubation with either rabbit or mouse horseradish peroxidase-conjugated secondary immunoglobulin (Jackson ImmunoResearch, West Grove, PA) at room temperature. Membranes were washed 4 x 10 min prior to detection of immune complexes by enhanced

chemiluminescence (Bio-Rad). Images were acquired using a ChemiDoc Touch imaging system (Bio-Rad), and densitometric analysis was performed using ImageLab software (Bio-Rad).

### Adenovirus infection

Cells grown to ~70% confluency were infected with Ad5-IκBαΔN or Ad5-GFP adenovirus at MOI 25 in serum-containing medium, as previously described [43]. After 24 h, the cells were incubated in fresh serum-free media overnight prior to treatments.

### Short interfering RNA-mediated knockdown

Pools of four non-targeting siRNAs (SI03650325, SI03650318, SI04380467, 1022064), GR siRNA (SI00003745, SI00003766, SI2654757, SI02654764), or RELA siRNA (SI02663101, SI05146204, SI00301672, SI02663094) (all Qiagen, Mississauga, ON, Canada) were mixed with 3 μL lipofectamine RNAiMax (Thermo Fisher Scientific) in 100 μl Opti-MEM (Thermo Fisher Scientific) and then incubated at room temperature for 5 min. This mixture was then added to A549 cells that were grown to ~70% confluency in serum-containing growth medium and incubated for 24 to 48 h until cells were >90% confluent.

### Data presentation and statistical analyses

GraphPad Prism 9 software (Graph-Pad Software, San Diego, CA) was used for generating data figures and performing statistical analyses. Data are plotted as bar or line graphs showing means ± S.E. of '*N*' independent observations. For normally distributed data, multiple comparison between groups were made by one-way ANOVA, with post-hoc tests (Dunnett, Bonferroni or Tukey), as indicated. Equivalent nonparametric tests were utilized for non-normal distribution. Two-tailed, paired Student's *t* tests were used for comparing two treatment groups. To test for greater than simple additivity, the sum of the effects (i.e. fold -1 or stimulated level–basal level) of each treatment was compared to the effect of cotreatment. Competitive antagonism at GR was shown by Schild analysis to produce $pA_2$ values, defined as the negative log of the molar concentration of antagonist necessary to double the concentration of agonist required to achieve the original response, as previously described [44].

## Results

### Effect of proinflammatory cytokines and glucocorticoids on BIRC2 and BIRC3 mRNA expression

To characterize the expression of BIRC2 and BIRC3, IL1B and TNF were used as inflammatory stimuli and dexamethasone or budesonide as representative glucocorticoids. A549 cells harboring an NF-κB luciferase reporter were first treated with various concentrations of IL1B and TNF to establish the maximum response to each cytokine. IL1B at 1 ng/ml and TNF at 10 ng/ml maximally activated NF-κB-dependent transcription (**S1A and S1B Fig in S1 File**). This effect was consistent with the induction of NF-κB DNA binding and these concentrations were used for all subsequent experiments [45].

BIRC2 and BIRC3 mRNA expression was significantly increased by IL1B and TNF with peak expression apparent at 4 h (**Fig 1A**). Thereafter, BIRC2 and BIRC3 mRNA expression sharply decreased in the IL1B-treated cells compared to the more gradual decreases in TNF-treated cells. Dexamethasone modestly, but significantly, increased BIRC3 mRNA expression, whereas BIRC2 mRNA was unaffected (**Fig 1A, S2A Fig in S1 File**). Increased BIRC3, but not BIRC2, mRNA expression also occurred in cells treated with the clinically relevant glucocorticoid, budesonide (**S2A Fig in S1 File**). Compared to IL1B treatment alone, IL1B-plus-

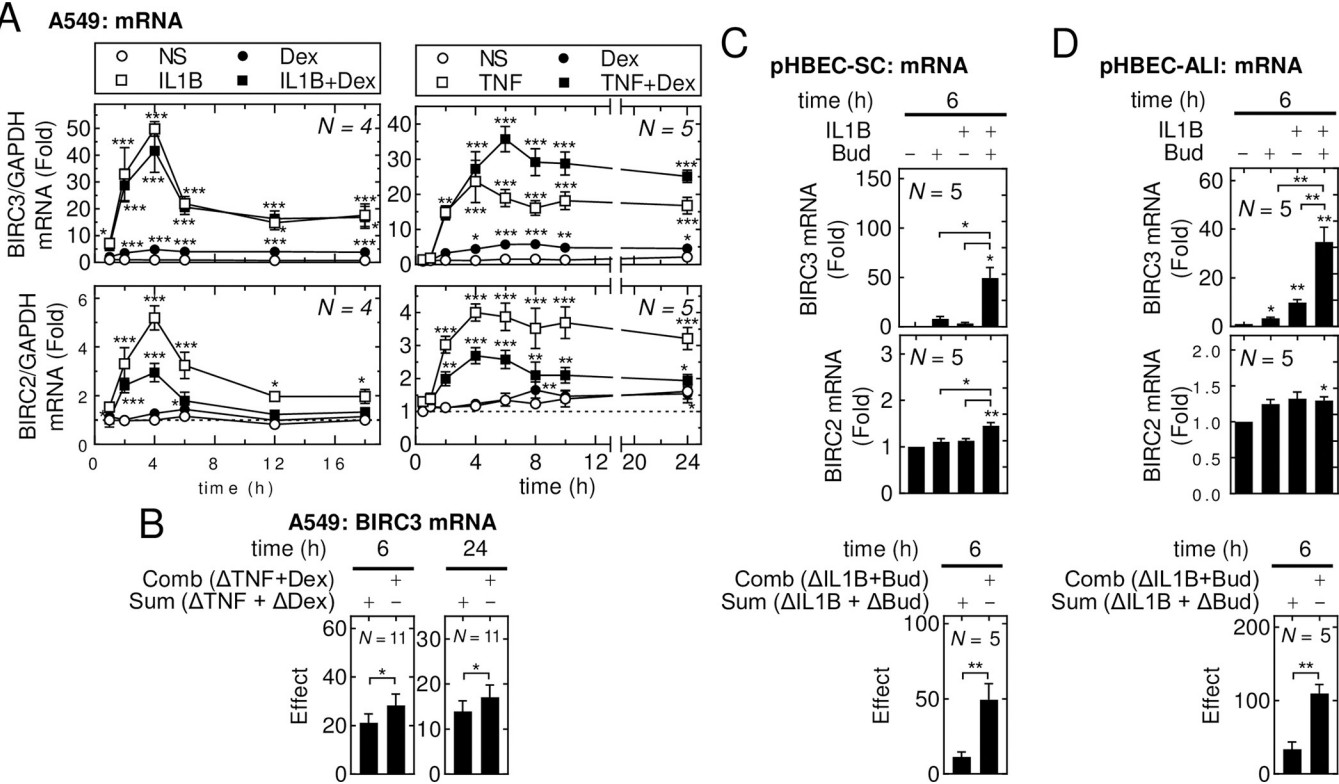

**Fig 1. Inflammatory cytokines and glucocorticoids differentially regulate BIRC2 and BIRC3 mRNA expression. (A-B)** A549 cells, **(C)** pHBEC-SC or **(D)** pHBEC-ALI were not stimulated (NS) or treated with IL1B (1 ng/ml), TNF (10 ng/ml), and/or dexamethasone (*Dex*; 1 μM) or budesonide (*Bud*; 300 nM). Cells from $N$ = 4–5 experiments were harvested at the indicated times for qPCR analysis. **(B)** The sum of the effects (*i.e.*, fold– 1) for TNF and Dex (*sum(ΔTNF + ΔDex)*) and the effect (fold– 1) of the combination treatment (*comboΔ(TNF + Dex)*) at 6 and 24 h is plotted. **(C & D)** The sum of the effects of IL1B and Bud (*sum(ΔIL1B + ΔBud)*) and the effect (fold– 1) of the combination treatment (*comboΔ(IL1B+Bud)*) at 6 h is plotted. Data for BIRC3 and BIRC2 were normalized to GAPDH, expressed as fold of NS either at t = 1 h (A), or at 6 h (C & D) and are plotted as mean ±SE. Significance was tested using one-way ANOVA with a Dunnett's post-hoc test in *A*, and a Tukey's post-hoc test in *C & E*. Significance in *B*, *C* and *D* was tested by paired *t* test. * $p \leq 0.05$, ** $p \leq 0.01$, *** $p \leq 0.001$ indicates significance relative to NS or as otherwise indicated.

dexamethasone showed no change in BIRC3 mRNA, whereas IL1B-induced BIRC2 mRNA was markedly and significantly repressed by dexamethasone (**Fig 1A and S2B Fig in S1 File**). While TNF-induced BIRC2 mRNA was also reduced by dexamethasone, TNF-induced BIRC3 mRNA was further enhanced by dexamethasone at all timepoints after 4 h (**Fig 1A**). These effects were significant for both genes at 6 and 24 h (**S2C Fig in S1 File**). Indeed, the combination effect for TNF-plus-dexamethasone was greater than the sum of the effects of each individual treatment and this confirms *supra*-additivity (**Fig 1B**). Since BIRC genes may play roles in cell fate following viral infection [27], we tested whether the viral infection-associated cytokine, IFNG, could enhance BIRC2 and/or BIRC3 expression. However, IFNG, at 10 ng/ml, alone or in the presence of IL1B or/and dexamethasone did not materially change BIRC2 or BIRC3 mRNA expression at 6 h (**S2D Fig in S1 File**). A similar lack of effect for BIRC2 was apparent at 24 h, whereas BIRC3 mRNA induced by IL1B, or IL1B-plus-dexamethasone was modestly enhanced by IFNG.

To explore the utility of A549 cells as a model, the expression BIRC2 and BIRC3 was examined in pHBEC grown as both submersion culture (pHBEC-SC) and as highly differentiated cells at air-liquid interface (pHBEC-ALI) [41]. In pHBEC-SC, BIRC3 mRNA expression was significantly increased by IL1B at 2 h, whereas BIRC2 was unaffected (**S2E Fig in S1 File**). Similar, but non-significant effects occurred at 6 h and at both times budesonide modestly induced

BIRC3 mRNA expression and BIRC2 mRNA was unaffected (**S2E Fig in S1 File and Fig 1C**). Treatment of pHBEC-SC with IL1B-plus-budesonide significantly increased BIRC3 mRNA expression at both 2 and 6 h and this effect was enhanced relative to the combined effects of the IL1B and budesonide treatments alone (**S2E Fig in S1 File and Fig 1C**). BIRC2 mRNA expression was also modestly enhanced by IL1B-plus-budesonide at 6 h (**Fig 1C**). IFNG had little effect on BIRC2 or BIRC3 mRNA expression alone or in combination with IL1B or/and budesonide (**S2E Fig in S1 File**). Using ALI culture of pHBECs, IL1B and budesonide each significantly enhanced BIRC3 mRNA expression at 6 h (**Fig 1D**). IL1B-plus-budesonide markedly enhanced BIRC3 mRNA expression and this was significantly greater than with each treatment alone. Indeed, the effect of the combination was significantly greater than the sum of the effects of each treatment and this supports *supra*-additivity at 6 h (**Fig 1D**). In pHBEC-ALI, BIRC2 mRNA expression was not significantly affected by either IL1B or budesonide but was modestly and significantly enhanced by the combination of IL1B plus budesonide.

Taken together, these results show that in A549 cells IL1B highly induced BIRC3 mRNA with no additional effect of glucocorticoid. Similarly, TNF also induced BIRC3 mRNA, but revealed cooperation with glucocorticoids to further enhance BIRC3 expression at all times from 6 h. In contrast, BIRC2 mRNA expression was only modestly enhanced by IL1B, or TNF, and this was reduced by glucocorticoid. Although similar data were obtained from primary epithelial cells, as pHBEC-ALI or -SC, greater cooperativity with glucocorticoid was apparent for BIRC3. In the pHBECs, BIRC2 was generally less induced, but showed modest enhancement by combinations with glucocorticoid. In both A549 cells and pHBECs, IFNG had no effect on BIRC2 or BIRC3 mRNA expression at 6 h, but promoted some enhancements at 24 h. Many of these later effects were more variable for BIRC3 in the pHBECs. Since events occurring at 24 h were likely due to secondary effects of IFNG stimulation, these were not further studied.

## Effect of proinflammatory cytokines and glucocorticoids on BIRC2 and BIRC3 protein expression

Protein expression for BIRC2 and BIRC3 was also characterized in multiple models of pulmonary epithelial cells to identify possible commonalities and/or differences with primary cells. Protein expression for BIRC2 and BIRC3 was investigated in A549 cells treated with IL1B, TNF and/or dexamethasone, or budesonide (**S3 Fig in S1 File**). Peak, or near peak, BIRC3 protein was observed from 6 h following IL1B and TNF treatments. Dexamethasone, or budesonide alone produced gradual, and only modest, increases in BIRC3 protein expression with significance reached at 24 h (**S3 Fig in S1 File**). IL1B-plus-dexamethasone or TNF-plus-dexamethasone treated cells also showed robust and significantly enhanced BIRC3 protein expression from 6–24 h, with greatest effects occurring at 24 h (**S3 Fig in S1 File**). BIRC2 protein expression was largely unaffected by IL1B, TNF and/or dexamethasone treatments (**S3 Fig in S1 File**). At 6 and 24 h, BIRC3 protein expression was significantly enhanced by IL1B and dexamethasone, with the effect of IL1B-plus-dexamehthasone combination was comparable to IL1B alone (**Fig 2A**). TNF significantly increased BIRC3 protein levels and this was further enhanced with TNF-plus-dexamethasone at 24 h (**Fig 2A**). Comparing the sum of the effects of TNF and dexamethasone alone with the effect of TNF-plus-dexamethasone indicated *supra*-additivity (**Fig 2B**). BIRC2 protein expression was not significantly affected by any of these treatments.

A549 cells were treated for 6 h with various concentrations of IL1B, TNF, dexamethasone, or budesonide and BIRC3 protein expression was evaluated by western blotting. In each case, IL1B, TNF, dexamethasone, and budesonide produced concentration-dependent enhancements of

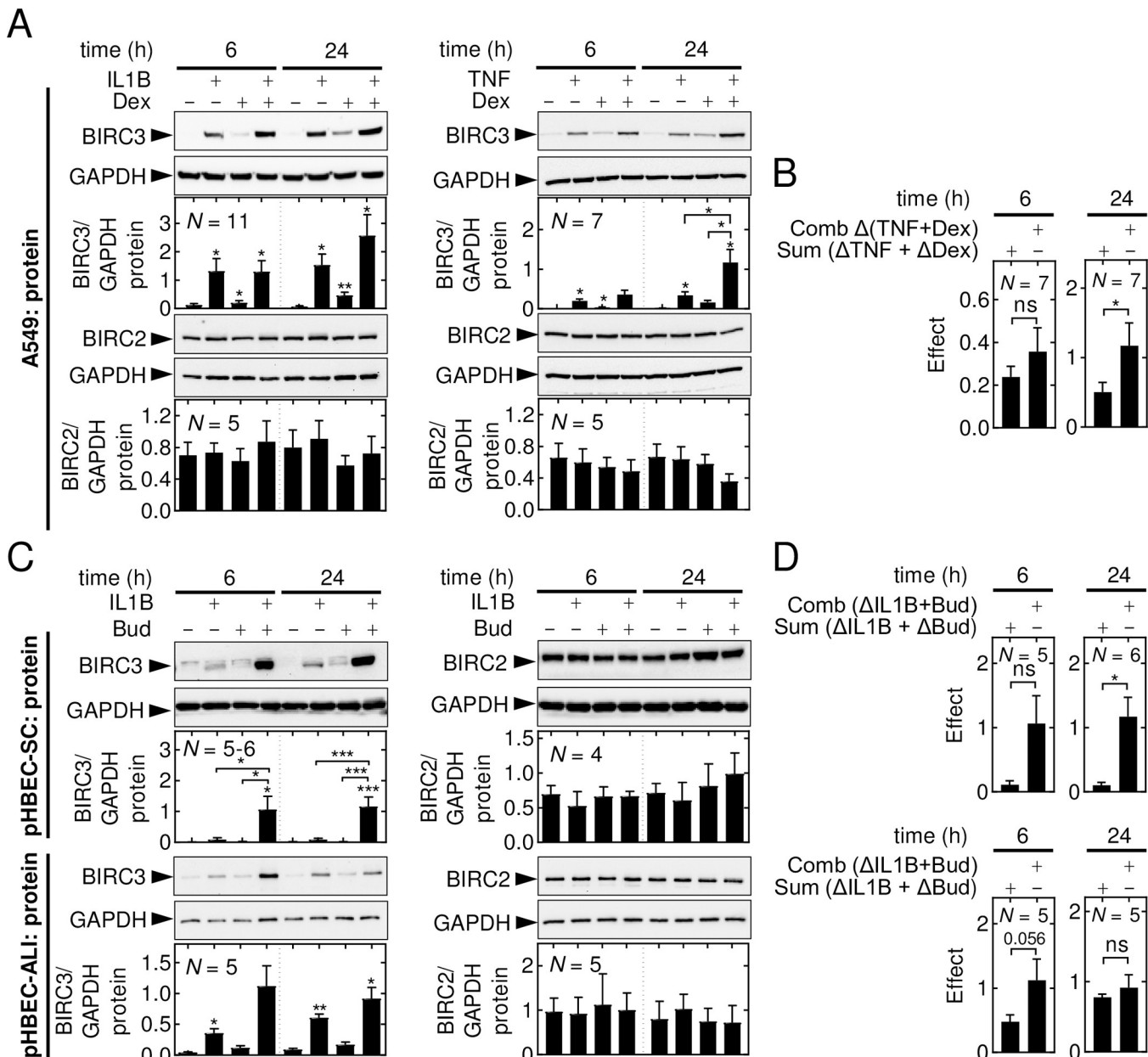

**Fig 2. Inflammatory cytokines and glucocorticoids differentially regulate BIRC2 and BIRC3 protein expression. (A-B)** A549 cells, **(C-D)** pHBEC-SC or pHBEC-ALI were not stimulated (NS) or treated with IL1B (1 ng/ml), TNF (10 ng/ml), and/or dexamethasone (*Dex*; 1 μM) or budesonide (*Bud*; 300 nM). Cells were harvested at 6 and 24 h for western blot analysis. **(B)** The sum of the effects (*i.e.*, fold– 1) for TNF and Dex (*sum(ΔTNF + ΔDex)*) and the effect (fold– 1) of the combination treatment (*comboΔ(TNF + Dex)*) from A, is plotted. **(D)** The sum of the effects (*i.e.*, fold– 1) of IL1B and Bud (*sum(ΔIL1B + ΔBud)*) and the effect (fold– 1) of the combination treatment (*comboΔ(IL1B+Bud)*) from C, is plotted. Blots representative of N = 4–11 experiments are shown. Data for BIRC3 and BIRC2 were normalized to GAPDH and plotted as mean ±SE. Significance was tested using one-way ANOVA with a Tukey's post-hoc test in *A* and *C*. Significance in *B* and *D* was tested by paired *t* test * $p \leq 0.05$, ** $p \leq 0.01$, *** $p \leq 0.001$ indicates significance relative to NS or as otherwise indicated.

BIRC3 expression with EC$_{50}$ values of 51.3 pg/ml, 4.3 ng/ml, 1.8 nM and 3.8 nM respectively (**S4 Fig in S1 File**). This confirmed that maximal, or near-maximal, concentrations of each cytokine and glucocorticoid were used in the prior experiments. Thus, the observed *supra*-additivity between cytokines and glucocorticoids on BIRC3 expression represents positive co-operativity between pathways that were maximally activated by each treatment.

In bronchial epithelial BEAS-2B cells, IL1B and dexamethasone at 6 h each modestly enhanced BIRC3 protein expression and this was significantly enhanced by the combination treatment (**S5A Fig in S1 File**). Similar, although non-significant, effects were apparent at 24 h. When compared to IL1B or dexamethasone alone, the combination of IL1B-plus-dexamethasone was greater than the sum of the effects at 6 h, showing *supra*-additivity, but not at 24 h (**S5B Fig in S1 File**). In Calu-3 cells, an immortalized epithelial cell line derived from human bronchial submucosal glands, IL1B significantly increased BIRC3 protein with a more modest increase produced by dexamethasone alone (**S5A Fig in S1 File**). However, while BIRC3 protein expression was significantly enhanced by the combination treatment and this was significantly greater than for IL1B or dexamethasone alone, the effect was not significantly more than additivity (**S5B Fig in S1 File**). pHBEC-SC showed only minor enhancements in BIRC3 protein expression with IL1B or budesonide but showed considerably greater and significant enhancements with IL1B-plus-budesoinde at 6 and 24 h (**Fig 2C**). Numerically, this was *supra*-additive at both timepoints, but was only significant at 24 h (**Fig 2D**). In pHBEC-ALI culture, IL1B induced BIRC3 expression at both 6 and 24 h, whereas the effect of budesonide was more modest and not significant (**Fig 2C**). IL1B-plus-budesonide significantly induced BIRC3 expression at 24 h. While not significant, the enhancement of BIRC3 protein by the combination was numerically *supra*-additive (*p*-value 0.056) at 6 h (**Fig 2D**). In BEAS-2B, Calu-3 and pHBEC-SC or pHBEC-ALI's, BIRC2 expression was largely unaffected by IL1B and/or dexamethasone/budesonide.

The above findings highlight many similarities in the expression of BIRC2 and BIRC3 protein when induced by inflammatory cytokines and/or glucocorticoids in undifferentiated and highly differentiated primary cell culture as well as in cell lines. While there may be cooperative interactions occurring between inflammatory stimulus and glucocorticoids in some models, the data nevertheless support the use of A549 cells to investigate the acute mechanisms of regulation observed for each stimulus. In the following analyses, we investigated regulation primarily at 6 h to avoid indirect effects that would prove difficult to delineate mechanistically.

## Role of NF-κB in the induction of BIRC2 and BIRC3 expression by proinflammatory cytokines

To explore contributions due to the NF-κB pathway in the ability of IL1B or TNF to induce BIRC2 and BIRC3 expression, pools of 4 control- or RELA-targeting siRNAs were titrated against RELA expression in NF-κB luciferase reporter cells (**Fig 3A**). The control siRNA pool was without effect, while maximal reductions in RELA expression were achieved by the RELA-targeting pool at 1 and 10 nM. This correlated with dramatic losses of IL1B-induced NF-κB luciferase reporter activity. In the presence of IL1B or TNF, silencing of RELA significantly reduced both IL1B- and TNF-induced NF-κB reporter activity (**Fig 3B**), these data are consistent with key roles for RELA in the induction of NF-κB-dependent transcription leading to IL1B- and TNF-induced BIRC3. Furthermore, the RELA-targeting siRNAs significantly reduced TNF-, IL1B- and IL1B-plus-dexamethasone-induced BIRC3 protein expression (**Fig 3C and S6 Fig in S1 File**). The loss of BIRC3 protein, that was apparent in unstimulated cells following RELA-targeting (**Fig 3C**), raises a possible role for basal NF-κB activity in maintaining low-level BIRC3 protein expression. Any changes in BIRC2 protein expression were only modest and not significant (data not shown).

In A549 cells, over-expression of the dominant inhibitor, IκBαΔN, is highly effective at preventing IL1B-induced activation of NF-κB [43]. In prior studies [46], A549 cells were infected (MOI 25) with Ad5-IκBαΔN or, as a control, Ad5-GFP, followed by treatment with IL1B or IL1B-plus-budesonide for 24 h. While Ad5-IκBαΔN robustly prevented NF-κB-dependent

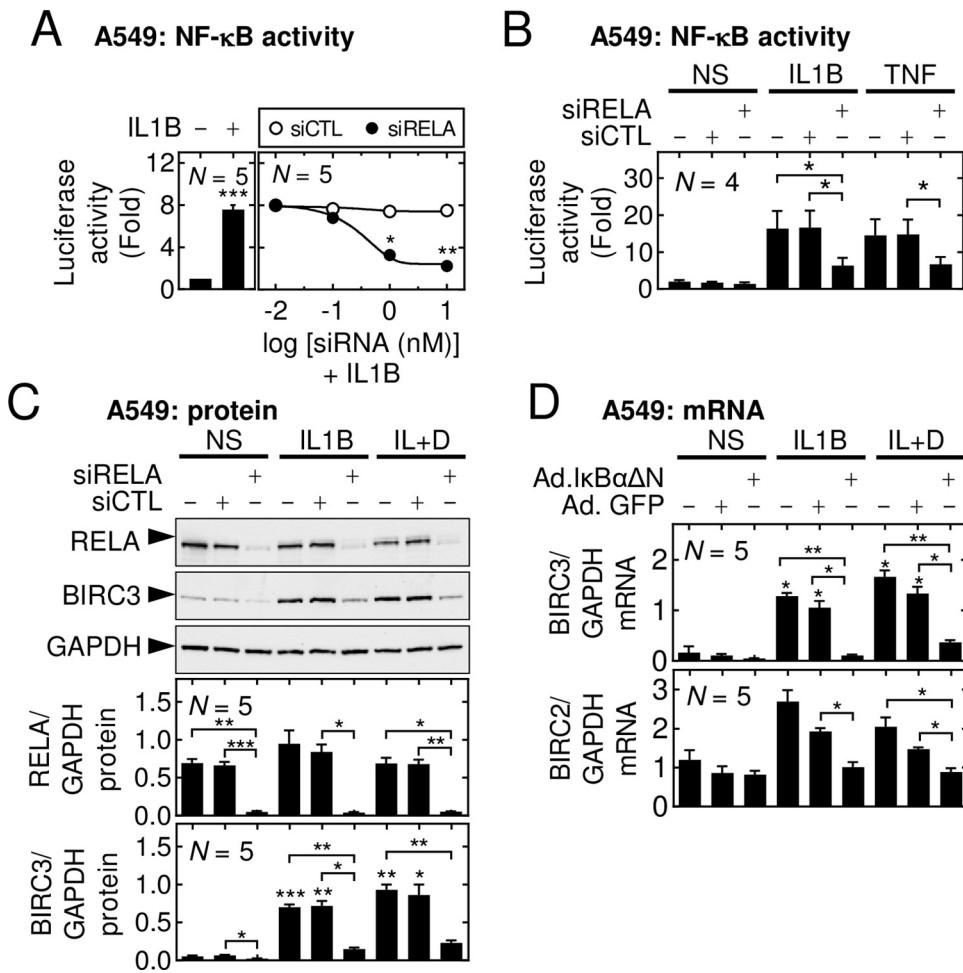

**Fig 3. Effect of NF-κB inhibition on BIRC2 and BIRC3 expression induced by proinflammatory cytokines. (A)** A549 cells harbouring the NF-κB-dependent reporter, 6κBtkluc.neo, were not stimulated (NS) or treated with IL1B (1 ng/ml), with or without prior incubation with increasing concentrations of control siRNAs (siCTL) or RELA-targeting siRNAs (siRELA) for 48 h (upper panel). **(B)** NF-κB-dependent reporter cells were not stimulated (NS) or treated with IL1B (1 ng/ml), or TNF (10 ng/ml), with or without prior incubation with siCTL or siRELA (both at 1 nM) for 48 h. After 6 h, cells were harvested for luciferase activity determination and data are plotted as fold of NS ±SE. **(C)** As in *A*, A549 cells were incubated with or without siCTL or siRELA, each at 1 nM, prior to stimulation with IL1B (1 ng/ml) or IL1B-plus-dexamethasone at 1 μM (IL+D). Cells were harvested at 6 h for western blot analysis of BIRC3, BIRC2 (data not shown), RELA and GAPDH. Representative blots are shown. **(D)** A549 cells were either not infected, or infected with Ad5-IκBαΔN or Ad5-GFP at an MOI of 25, before treatment with IL1B or IL+D. Cells were harvested at 6 h for qPCR analysis of BIRC2, BIRC3 and GAPDH. In *C* and *D*, mRNA or protein data for BIRC3, BIRC2 or RELA were normalized to GAPDH and are plotted as mean ± SE. Data from $N$ = 4–5 experiments are shown, and significance was tested in panels *A*, *C* and *D* using one-way ANOVA with a Tukey's post-hoc test. Significance in *B* was tested by paired *t* test. * $p \leq 0.05$, ** $p \leq 0.01$, *** $p \leq 0.001$ indicates significance relative to NS or as otherwise indicated.

transcription [46], there was little effect on basal BIRC3 mRNA expression, but IL1B- or IL1B-plus-dexamethasone-induced BIRC3 mRNA was abolished (**Fig 3D**). Additionally, the modest induction of BIRC2 mRNA in IL1B- and IL1B-plus-dexamethasone-treated cells was prevented by Ad5-IκBαΔN, whereas basal BIRC2 mRNA expression was unaffected by either Ad5-IκBαΔN or Ad5-GFP. Similarly, the selective IKK2 inhibitor, PS-1145, reduced activation of NF-κB [47,48] and, at 30 μM maximally inhibits IκBα (serine 32 and 36) phosphorylation and NF-κB-dependent transcription [49]. This concentration of PS-1145 significantly decreased BIRC3 mRNA expression that was induced by IL1B and IL1B-plus-dexamethasone

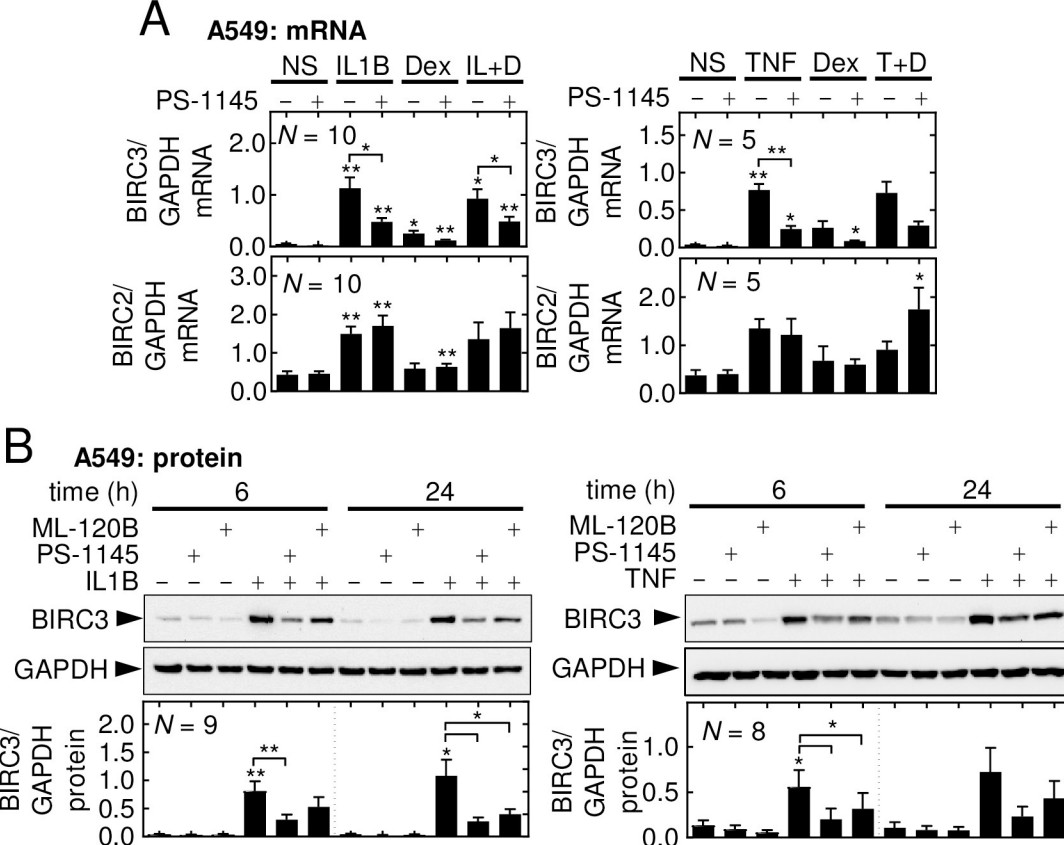

**Fig 4. Effect of IKK2 inhibitors on BIRC2 and BIRC3 expression. (A)** A549 cells were pretreated, or not, with PS-1145 (30 μM) for 90 min prior to addition of IL1B (1 ng/ml), TNF (10 ng/ml), dexamethasone (*Dex*; 1 μM) or in combination (IL+D, T+D), as indicated. Cells were harvested at 6 h for qPCR analysis. **(B)** A549 cells were pretreated, or not, with PS-1145 (30 μM) or ML-120B (30 μM) for 90 min prior to treatment with IL1B (1 ng/ml) or TNF (10 ng/ml). Cells were harvested at 6 and 24 h for western blot analysis. Representative blots are shown. *A & B*, BIRC2 and BIRC3 mRNA or protein (BIRC2 data not shown) data, from $N$ = 5–10 experiments, were normalized to GAPDH and plotted as mean ±SE. Significance was tested using one-way ANOVA with a Tukey's post-hoc test in *A*, and Bonferroni's post-hoc test in *B*. * $p \leq 0.05$, ** $p \leq 0.01$ and *** $p \leq 0.001$ indicates significance relative to NS or as otherwise indicated.

(**Fig 4A**). TNF-induced BIRC3 mRNA expression was also significantly reduced by PS-1145 and in TNF-plus-dexamethasone treated cells, BIRC3 mRNA was markedly reduced, although this did not reach significance (**Fig 4A**). Dexamethasone-induced BIRC3 mRNA, while not significantly changed by PS-1145, was noticeably reduced and this is consistent with the repressive effect of IκBαDN on basal BIRC3 expression in the prior experiment. Following IL1B or TNF treatments, with or without dexamethasone, BIRC2 mRNA expression was unaffected by PS-1145 (**Fig 4A**). A549 cells were also pre-treated with PS-1145 and the related compound, ML-120B, which is maximally effective in A549 cells at 30 μM [49]. PS-1145 significantly reduced IL1B-induced BIRC3 protein expression at 6 and 24 h (**Fig 4B**). Significant loss of BIRC3 protein was also shown for PS-1145 on the TNF treatment at 24 h. Likewise, ML-120B significantly reduced BIRC3 expression in cells treated with IL1B at 24 h or TNF at 6 h. Expression of BIRC2 protein was modestly, and with the exception of ML120B on the TNF treatment at 24 h, not significantly, reduced by PS-1145 and ML-120B in IL1B or TNF-treated cells (data not shown). Taken together, these data suggest that IKK2, in the canonical NF-κB activation pathway, contributes to IL1B- and TNF-induced BIRC3 gene expression. However, the lack of effect of PS-1145 on IL1B or TNF-induced BIRC2 mRNA, yet repression by

IκBαDN, raises the possibility of BIRC2 induction via IKK2-independent, non-canonical pathways.

## BIRC3 expression induced by glucocorticoids requires GR

Possible roles for GR in the ability of glucocorticoid to induce BIRC3 expression were evaluated using ORG34517, a competitive GR antagonist [44,50], and siRNA-mediated silencing of GR. In A549 cells, BIRC3 mRNA expression was induced by dexamethasone and budesonide in a concentration-dependent manner that, as shown by the right-shifts in their response curves, was competitive with ORG34517 (**Fig 5A** and **S7A Fig** in **S1 File**). Schild analysis produced $pA_2$ values of 8.4 and 8.0 for dexamethasone and budesonide, respectively, which is consistent with competition at GR [44]. Similarly, ORG34517 significantly antagonized the ability

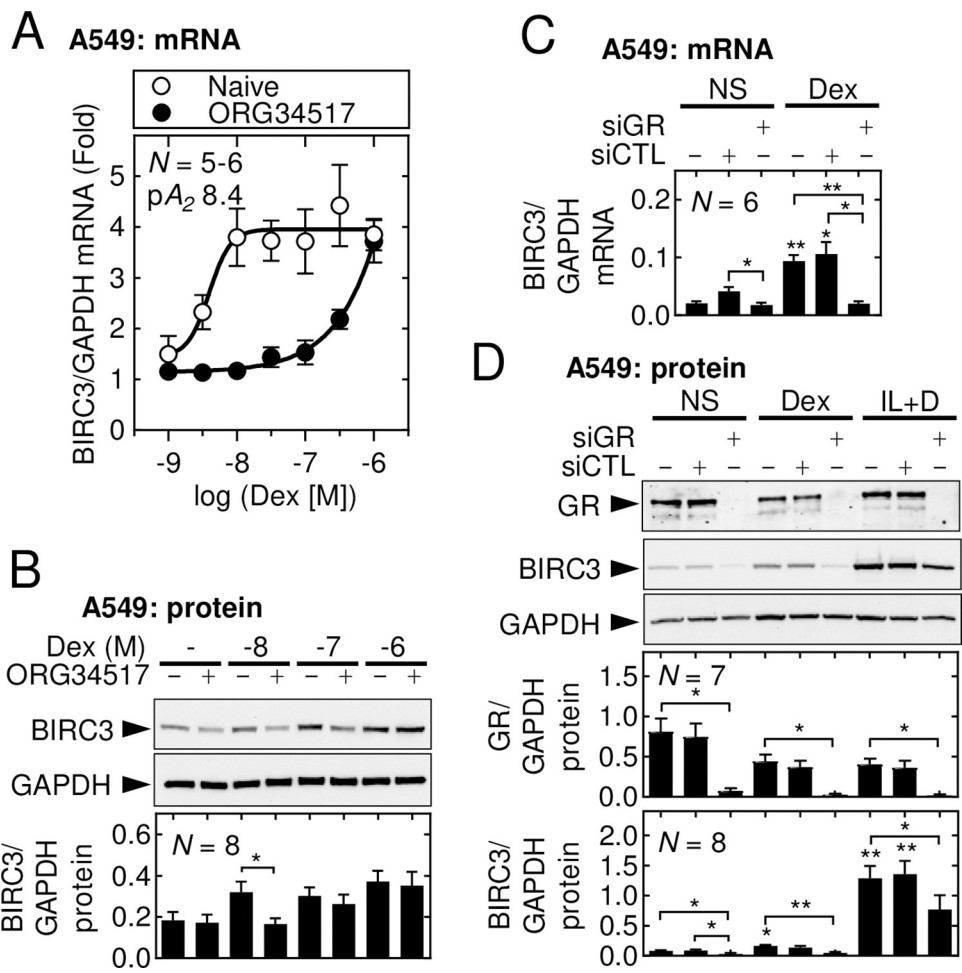

**Fig 5. Glucocorticoid-induced expression of BIRC3 requires GR.** (A-B) A549 cells were either not treated or pre-treated with ORG35417 (1 μM) for 1 h before addition of dexamethasone (Dex) at the indicated concentrations. Cells were harvested at 6 h for; *A*, qPCR and Schild analysis, or *B*, western blot analysis. (C-D) A549 cells were incubated with either control siRNAs (siCTL) or GR-targeting siRNAs (siGR) for 48 h prior to addition of dexamethasone (*Dex*; 1 μM), IL1B (1 ng/ml), or IL1B plus dexamethasone (IL+D). Cells were harvested at 6 h for western blot analysis of BIRC3, GR and GAPDH. Representative blots are shown. BIRC3 and GR mRNA or protein data, from *N* = 5–8 experiments, were normalized to GAPDH and plotted as mean ±SE. Significance was tested using one-way ANOVA with a Tukey's post-hoc test. * $p \leq 0.05$ and ** $p \leq 0.01$ indicates significance relative to NS or as otherwise indicated.

of dexamethasone to induce BIRC3 protein at low concentrations but was ineffective against dexamethasone at 1 μM (**Fig 5B**).

In A549 cells harboring a stably integrated 2×GRE luciferase reporter, each of four GR-targeting siRNAs strongly suppressed GR protein expression and significantly reduced 2×GRE reporter activity induced by dexamethasone, whereas control siRNA had no effect (**S7B Fig in S1 File**). Pooling of the four siRNAs to achieve the same aggregate final concentration yielded an essentially identical inhibition curve (data not shown) and was subsequently used at final concentration of 1 nM. Following dexamethasone treatment, the GR-targeting siRNA pool, but not the control siRNAs, strongly and significantly reduced BIRC3 mRNA and protein expression (**Fig 5C and 5D**). GR silencing was also confirmed. In addition, BIRC3 protein was significantly reduced following GR targeting in the IL1B-plus-dexamethasone treated cells (**Fig 5D**). Thus, glucocorticoid enhanced and IL1B plus glucocorticoid-induced BIRC3 expression appears to require the GR.

## TNF selectively destabilizes BIRC2 and BIRC3 protein

To investigate the stability of BIRC2 and BIRC3 proteins, A549 cells were pre-treated with IL1B or TNF for 1 h prior to addition (t = 0) of the translational blocker, cycloheximide (CHX), and then harvested at the indicated timepoints (**Fig 6A**). This protocol was adopted to allow possible effects on BIRC expression to be assessed while reducing the impact of protein synthesis inhibition on the control of acutely-activated signaling pathways [45,51]. After 1 h (i.e. t = 0), there was relatively little effect of IL1B or TNF on either BIRC3 or 2 expression (**Fig 6A**). Nevertheless, 1 h of TNF (t = 0) significantly reduced BIRC2 expression and this is consistent with data depicted in S3 Fig in S1 File. IL1B and TNF markedly enhanced BIRC3 protein at t = 6 h (7 h post IL1B or TNF), while modestly, but not significantly, inducing BIRC2 protein expression (**Fig 6B**). In IL1B-plus-CHX-treated cells, BIRC3 and BIRC2 protein expression was not induced and even after 6 h, the expression remained similar to the level at t = 0 (**Fig 6B**). This was also apparent in unstimulated cells treated with CHX alone. However, 6 h following TNF-plus-CHX treatment, both BIRC3 and BIRC2 protein expression was consistently and significantly reduced compared to basal expression (**Fig 6B**). These data show divergent effects of IL1B and TNF on BIRC2 and BIRC3 protein stability, whereby TNF, but not IL1B, stimulation led to a dramatic loss of BIRC protein stability.

Given that BIRC3 protein expression induced by both IL1B and TNF was maximal, or near maximal, at 6 h (**Fig 2A**), A549 cells were stimulated with IL1B or TNF for 6 h prior to the addition of CHX (t = 0) (**Fig 7A**). In each case, IL1B and TNF strongly induced the expression of BIRC3 (i.e. at t = 0). In the absence of CHX, basal, IL1B- and TNF-induced BIRC3 remained constant for 6 h (**Fig 7B**). This was largely unaffected by the addition of CHX and there were no significant differences in BIRC3 expression between CHX, IL1B-plus-CHX or TNF-plus-CHX treatments. These data indicate that unlike basal expression of BIRC2 and BIRC3 proteins, following acute treatment with TNF, the BIRC3 protein expression that was induced by IL1B or TNF at 6 h remained stable and was relatively long-lived.

**Proteasome inhibition rescues TNF-induced degradation of BIRC2 and BIRC3.**   To further interrogate the role of protein degradation in regulating BIRC3 and 2 expression, A549 cells were incubated with MG-132, MG-262 or PR-171, which all inhibit the activity of the proteolytic 20S core of the 26S proteasome [52–54], and the cysteine proteinase inhibitor, E-64 [55]. Western blotting of whole A549 cell extracts for K48 ubiquitin revealed striking increases in signal following IL1B or TNF treatments in the presence of the three proteasome inhibitors, but not E-64 (**S8A Fig in S1 File**). As K48 ubiquitylation is typically the cue for proteolytic degradation via the 26S proteasome [56], these data suggest efficacy of each compound and

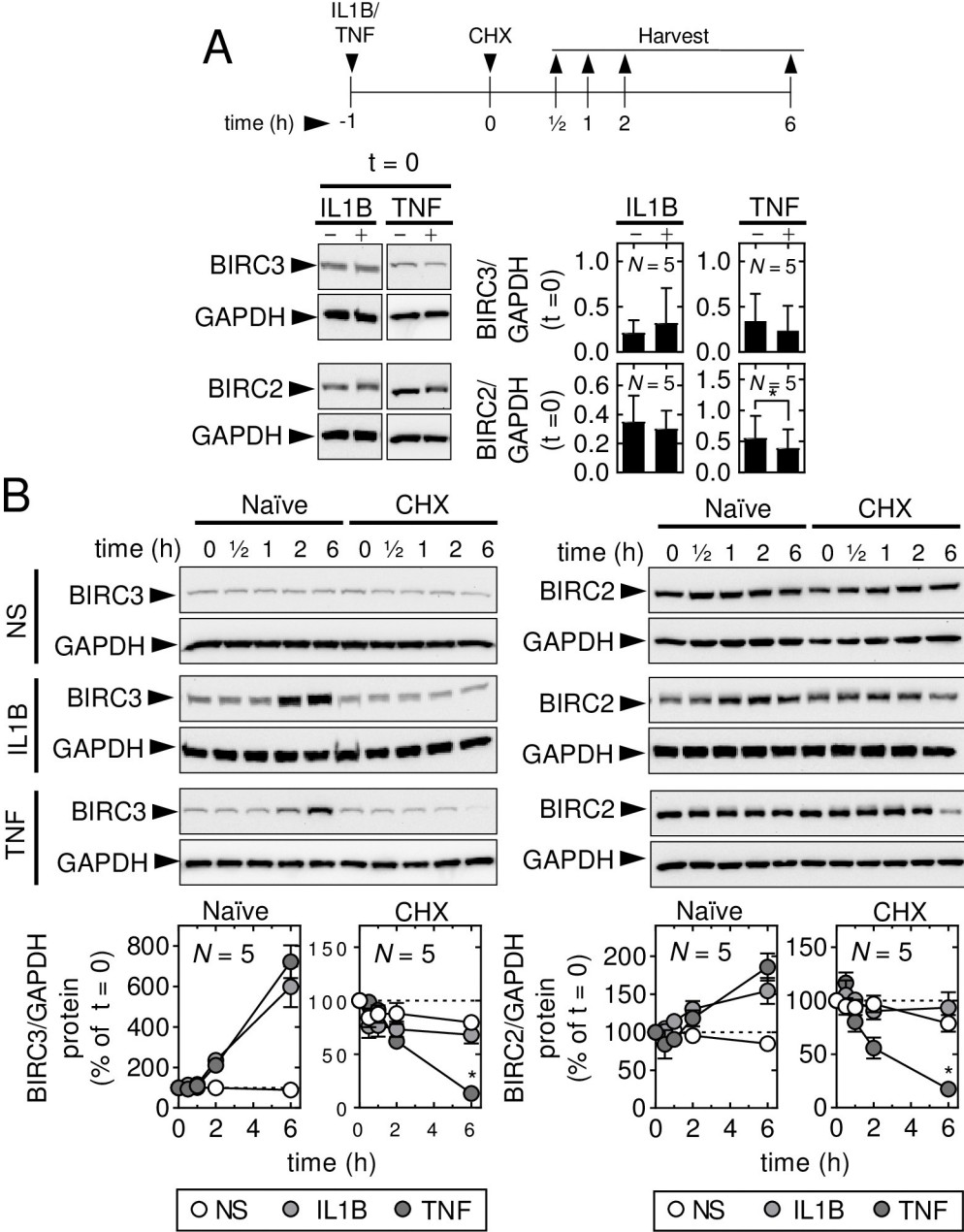

**Fig 6. TNF reduces BIRC2 and BIRC3 protein stability.** A549 cells were not stimulated (NS) or treated with IL1B (1 ng/ml), or TNF (10 ng/ml) for 1 h prior to the addition (t = 0) of cycloheximide (CHX; 10 µg/ml), as indicated. Cells were harvested: (**A**) at t = 0; and, (**B**) at t = 0, 0.5, 1, 2 and 6 h for western blot analysis of BIRC3, BIRC2 and GAPDH. Representative blots are shown for each panel. In *A*, densitometric data for BIRC2 and BIRC3 were normalized to GAPDH. In *B*, densitometric data, normalized to GAPDH, were expressed as a percentage of each treatment at t = 0. Data from N = 5 experiments are plotted as mean ±SE. Significance in *A*, was tested by paired *t* test and, in *B*, by one-way ANOVA with a Dunnett's post-hoc test. * $p \leq 0.05$ in *A* indicates significance relative to NS or in *B*, relative to NS (t = 0).

pave the way for analysis of BIRC expression following IL1B and TNF stimulation (**Fig 8A**). In unstimulated cells, there was no effect of the proteasome inhibitors or E-64 on basal BIRC3 or BIRC2 expression (**S8B Fig in S1 File and Fig 8B**). However, the induction of BIRC3 protein

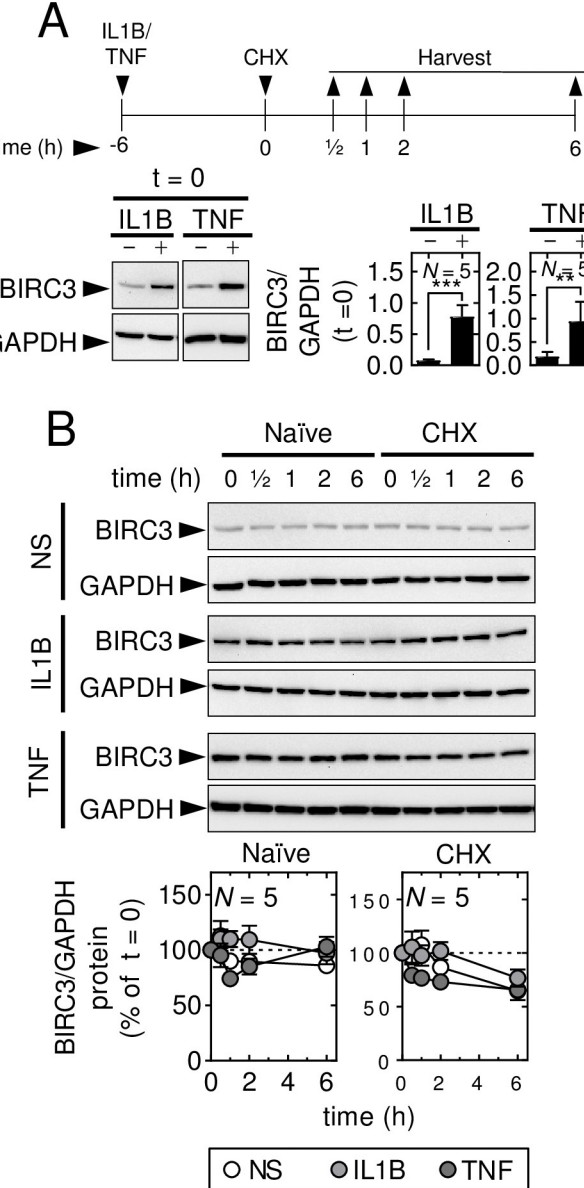

**Fig 7. IL1B- and TNF-induced BIRC3 protein is stable.** A549 cells were not stimulated (NS) or treated with IL1B (1 ng/ml), or TNF (10 ng/ml) for 6 h prior to the addition (t = 0) of cycloheximide (CHX; 10 µg/ml), as indicated. Cells were harvested: (**A**) at t = 0; and, (**B**) at t = 0, 0.5, 1, 2 and 6 h for western blot analysis of BIRC3 and GAPDH. Representative blots are shown for each panel. In *A*, densitometric data for BIRC3 were normalized to GAPDH. In *B*, densitometric data, normalized to GAPDH, were expressed as a percentage of each treatment at t = 0. Data from $N = 5$ experiments are plotted as mean ±SE. Significance in *A*, was tested by paired *t* test and, in *B*, by one-way ANOVA with a Dunnett's post-hoc test. ** $p \leq 0.01$ and *** $p \leq 0.001$ in *A* indicates significance relative to NS.

expression at 6 h by IL1B or TNF was prevented by MG-132, MG-262 and PR-171, but not E-64 (**Fig 8B**). Similar data were obtained 2 h following IL1B and TNF treatment (**S8B Fig in S1 File**). While proteolytic turnover of BIRCs may have been prevented by inhibitors of the 26S proteasome, the fact that both BIRC3 and 2 appear to be regulated via NF-κB (above) raises the prospect that inhibition of NF-κB was overriding and could therefore explain the current observations. Titration of each compound on an NF-κB-dependent luciferase reporter confirmed profound inhibition of IL1B and TNF-induced NF-κB activity by the proteosome

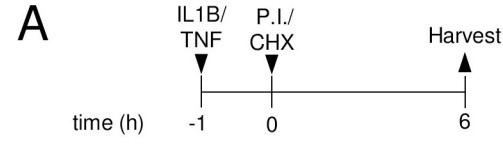

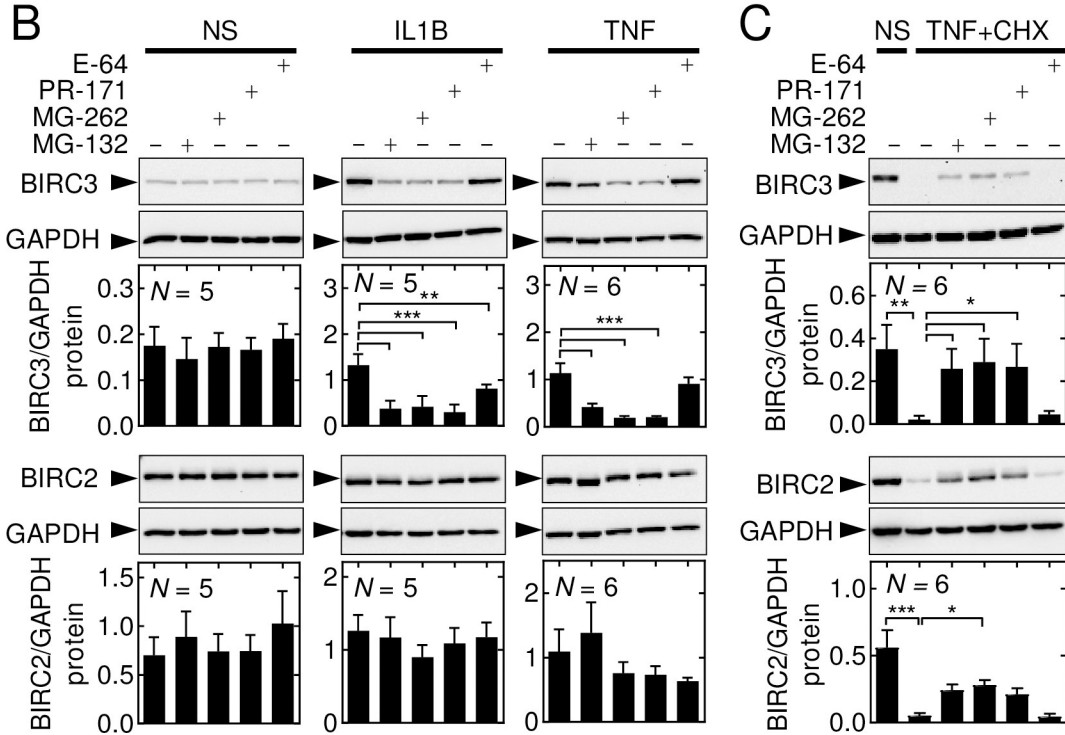

**Fig 8. Proteasome inhibition rescues TNF-induced loss of BIRC2 and BIRC3. (A)** As depicted in the schematic, A549 cells were either not stimulated (NS) or treated with IL1B (1 ng/ml), or TNF (10 ng/ml) for 1 h before the addition (t = 0) of MG-132 (10 μM), MG-262 (10 μM), PR-171 (10 μM) or E-64 (10 μM) either in the absence or presence of cycloheximide (CHX; 10 μg/ml). All cells were harvested at t = 6 h for western blot analysis. **(B)** Shows the effects of NS, IL1B and TNF and the inhibitors in the absence of CHX. **(C)** Shows NS or TNF in absence or presence of CHX along with the further effect of the inhibitors. Representative blots are shown and densitometric data for BIRC3 and BIRC2, normalized to GAPDH, and are plotted as mean ±SE. Data from N = 5–6 experiments are shown and significance tested using one-way ANOVA with a Tukey's post-hoc test. * $p \le 0.05$, ** $p \le 0.01$, *** $p \le 0.001$ indicates significance relative to NS or as otherwise indicated.

inhibitors, but a lesser effect of E-64 (**S8C Fig in** S1 File). A549 cells were therefore treated with TNF for 1 h prior to the addition of MG-132, MG-262, PR-171 and E-64 in the presence of CHX. Following TNF-plus-CHX treatment, BIRC3 expression was completely lost, but this was rescued by MG-132, MG-262 and PR-171, and not E-64 (**Fig 8C, upper panel**). Similarly, while BIRC2 expression was not modulated by TNF, in the presence of TNF-plus-CHX, BIRC2 expression virtually disappeared (**Fig 8C, lower**). The loss of BIRC2 in TNF-plus-CHX-treated cells was also markedly attenuated by MG-132, MG-262 and PR-171, but not by E-64 (**Fig 8C, lower panel**). These data confirm the TNF-induced loss of BIRC2 and BIRC3 proteins and indicate a role for the 26S proteasome in their degradation.

## Discussion

Precise roles for BIRC2 and BIRC3 remain controversial and they are commonly grouped together as crucial, but apparently redundant, positive regulators of TNF-induced NF-κB and

in resisting apoptosis [57]. Thus, BIRC2/3 double-deficient cells, or treatment with BIRC inhibitors, appears to reduce the ability to activate NF-κB and/or sensitizes to cell death [6,7,11]. However, while various reports also suggest that BIRC2 and BIRC3 may not be functionally interchangeable, such effects may be cell type, receptor and/or signal specific, and to date this remains poorly characterized [57–60]. Since the current data document clear differential regulation of BIRC3 and BIRC2 expression in multiple epithelial cell lines and pHBECs, considerable support is provided for the concept that there are key differences in the roles played by each proteins.

In the current studies, BIRC3 mRNA and protein were robustly induced by both IL1B and TNF in multiple pulmonary epithelial cell lines as well as in pHBECs grown as undifferentiated cells in submersion culture or as highly differentiated ALI cultures. Conversely, and despite potential roles for these two BIRCs in cell fate determination during viral infections [27], the antiviral cytokine, IFNG [61,62], did not acutely induce expression of either BIRC2 or BIRC3. While some effects of INFG were apparent on BIRC expression at 24 h, these were relatively modest, or variable, and a likely a consequence of IFNG inducing downstream effectors that then subsequently induced BIRC expression. Thus, the current data confirm a highly consistent response to inflammatory stimuli, such as IL1B or TNF, in lung epithelial cells that, at least for BIRC3 involved the transcription factor, NF-κB, and activation via an IKK2-dependent pathway. Roles for NF-κB are also supported by previous reports in T cells and promoter analyses using HEK 293 cells [63,64]. Furthermore, while low level expression of BIRC3 was detected in the absence of cytokine stimulation, and is therefore consistent with a possible direct, or immediate, role in signaling, the very considerable increases in BIRC3 expression following IL1B and TNF treatments suggest a more important role at later times following either treatment. These observations contrast markedly with the expression of BIRC2, which was readily detectible by western blotting in unstimulated cells but was relatively unaffected by longer durations of IL1B or TNF treatment. Thus, the near constitutive presence of BIRC2 protein is consistent with possible roles in rapid onset signaling events that occur without the need for new gene expression. However, both IL1B and TNF modestly induced BIRC2 mRNA expression, likely also via NF-κB-, or partially NF-κB-dependent mechanisms. Interestingly, the increases in BIRC2 mRNA induced by IL1B and TNF were refractory to IKK2 inhibitors and this raises the possibility of a non-canonical NF-κB pathway to induce BIRC2 expression. Overall, the relative lack of change in BIRC2 protein raises the possibility that BIRC2 undergoes signal-induced degradation but is then replaced via modestly increased *de novo* gene expression. Indeed, there was some initial loss of BIRC2 protein expression following TNF treatment that recovered over time. Furthermore, stimulation in the presence of cycloheximide confirmed that TNF, but not IL1B, reduced BIRC2 protein stability in a manner that was attenuated by inhibitors of the 26S proteasome. This effect of TNF was also apparent on basal BIRC3 protein present in unstimulated cells, whereas BIRC3 protein induced by IL1B, or TNF remained stable. Thus, the differential inducibility of BIRC3 compared to the more constitutive expression of BIRC2 suggests a later onset role for BIRC3, whereas BIRC2 would be readily available to participate in immediate signaling events. These current data therefore argue against a simple "redundant" role for these two proteins. Rather, roles in the rapid activation of NF-κB are most likely to be served by BIRC2, while the induction of BIRC3 expression may take on roles in later onset signaling events. The current data therefore provide a temporal framework to explore these effects.

The differential ability of TNF, but not IL1B, to induce proteolytic loss of BIRC2 and BIRC3 raises many further questions. Are these BIRCs both involved in signaling by IL1B and TNF? If so, why in the presence of TNF are there specific mechanisms that act to reduce BIRC expression (and presumably function)? Alternatively, is BIRC signaling only relevant in the

presence of TNF and is proteolytic loss a marker of prior BIRC activity? Equally, is the profound enhancement of BIRC3 expression by IL1B a mechanism to prime for anti-apoptotic effects to promote cell survival? While these questions do not currently have clear answers, most data appears to support a role for both BIRC2 and BIRC3 in responses to TNF [4,6,7,11,57]. As BIRC2 and BIRC3 proteins are suggested to be E3 ubiquitin ligases for both activation of signaling and protein degradation, there could be a coordinated switching on and off of TNFR-mediated signaling pathways based on differential BIRC expression kinetics. It is possible that BIRC2 acts as the initial E3 ligase intermediate to allow for TNFR-signaling that is either degraded or supplanted by BIRC3 for later onset signaling. Furthermore, TNF- and IL1B-induced BIRC3 protein was not subject to overt proteolytic degradation suggesting that once induced by NF-κB, the events that previously led to early degradation of BIRC2, or BIRC3, must no longer operate. Reasons for this are yet to be described, but it is salient to note that BIRC2 and BIRC3 are implicated in non-canonic signaling to NF-κB [16,57,65,66], as well as in responses from NODs [18], TLRs [17], other inflammatory receptors and pathways [20]. Differential roles for BIRC2 and BIRC3, and their possible degradation, will therefore need to be investigated in the context of these different receptor systems.

BIRC2 and BIRC3 also reveal striking differential regulation by glucocorticoids. Like many inflammatory genes that are regulated by NF-κB [31,67–69], IL1B- and TNF-induced BIRC2 mRNA was repressed by glucocorticoid co-treatment in A549 cells. However, this effect was not accompanied by corresponding reductions in BIRC2 protein expression suggesting that protein turnover might also be reduced leading to little overall effect. The functional significance of this remains to be explored. Effects of glucocorticoid on BIRC3 were possibly more surprising. In the presence of IL1B and TNF, which, as noted above, induce BIRC3 via an NF-κB-dependent mechanism, there was no evidence of repression by glucocorticoids. Rather in A549 cells, when BIRC3 expression was induced by IL1B it was unaltered by co-addition of glucocorticoid, whereas glucocorticoid in the presence of TNF produced a marked combinatorial, *supra*-additive effect. Similar failures to repress and synergies with a glucocorticoid were apparent in other epithelial cell lines and in both undifferentiated and differentiated pHBECs. Thus, among inflammatory genes induced by IL1B or TNF, BIRC3 stands out as one that is resistant to repression by glucocorticoids [31,70]. Indeed, the current data documents situations, including in primary epithelial cells, where there is synergy between the inflammatory stimulus and glucocorticoids on the induction of BIRC3 expression. Such observations may account for the presence of BIRC3 expression in severe asthma and the existence of BIRC3 polymorphisms that associate with asthma therefore warrant further investigation [26,71]. Mechanistically, BIRC3 expression was independently induced by glucocorticoids acting via the GR. Indeed, multiple GR bindings sites have been identified at the BIRC3 locus in response to glucocorticoids [35,72,73]. Thus, it is plausible that the presence of GR binding to conventional glucocorticoid response elements (GREs) could offset repression that occurs via other mechanisms [70]. Furthermore, the presence of GR and GREs at the *BIRC3* locus may enable transcriptional synergy with factors activated by IL1B or TNF. Positive interactions between GR and various factors, including NF-κB, are widely reported and may underpin the current observations [46,49,73–75]. Indeed, such interactions are proposed for a ~1 kb *BIRC3* promoter region upstream of transcription start that contains putative NF-κB and GRE sites [76]. However, it is salient to note that this previously reported region does not contain the main GR binding site identified by ChIP-seq [35,72,73], and it is therefore likely that important facets of this interaction remain to be described. Indeed, while the current analyses support roles for NF-κB and GR in the rapid induction of BIRC3 by IL1B, TNF and glucocorticoids, how these pathways then interact to produce synergistic increases in BIRC3 expression was not specifically addressed.

Functionally, the effects on BIRC3 expression warrant further consideration. Why would an anti-inflammatory glucocorticoid act to render BIRC3 resistant to repression, or to further increase its expression? We might speculate that it is possible that the induction of BIRC3 by inflammatory stimuli subserves functions that, if inhibited, are detrimental. Indeed, there are multiple examples of inflammatory genes that are maintained by glucocorticoids, for example to ensure feedback control of NF-κB, IL1B and TNF signaling, or inflammatory gene expression [70,74,75,77]. In the current context, the ability of glucocorticoid to maintain or enhance the expression of BIRC3 with concurrent pro-inflammatory stimuli could be important in maintaining or promoting anti-apoptotic effects, as has been previously suggested [76,78]. Such effects may be important in the context of viral infections, where for example during influenza infection, BIRC3 is thought to be protective [27].

In conclusion, the current data document differential regulation of BIRC2 and BIRC3 by inflammatory cytokines and glucocorticoids, which we summarize in **S1 Table**. We extend many prior observations into primary cells, including undifferentiated and highly differentiated pHBECs. These kinetic data support differential roles for each protein. Thus, the constitutive presence of BIRC2 is consistent with acute effects on signaling, whereas the profoundly induced expression of BIRC3 is more consistent with later onset roles. Importantly, BIRC3 expression was induced by glucocorticoids, and these acted to maintain or even enhance expression in the context of inflammatory stimuli. We therefore postulate that BIRC3 plays key roles in inflammatory signaling, including in the potential prevention of apoptosis. While this may have been conserved during evolution, the functional consequences of glucocorticoids on BIRC3 expression in the context of glucocorticoid therapy requires investigation.

## Supporting information

**S1 Table. Summary table of experimental findings.** Summary table highlighting the commonalities and differences between BIRC2 & BIRC3 expression and regulation found in this study. (PDF)

**S1 File. Expression of BIRC2 and BIRC3 in pulmonary epithelial cells.** A549, BEAS2-B, Calu-3, pHBEC-SC and pHBEC-ALI cells were treated with proinflammatory cytokines and glucocorticoids at specific times as per the in-text captions. Included are dose-response curves for IL1B and TNF, siRNA silencing for RELA and GR, proteasome inhibitors and the effect of cycloheximide on K48-Ub. Data for mRNA expression, protein expression and NF-κB- or GRE-dependent luciferase activity are shown. Each figure has a legend detailing the experimental parameters. (PDF)

**S2 File. Raw data and uncropped western blot images.** Included are tabular data values used in each individual graph of (1) mRNA expression represented as GENE/GAPDH, (2) NF-κB-dependent luciferase activity shown as relative light units (RLU) of Treatment RLU / Non-stimulated RLU and (3) protein expression as raw densitometric values given as GENE/GAPDH. Additionally, the original, uncropped images for each representative blot are shown with molecular weight marker included. Any lanes not included in the final figure have been marked with an "X" above each lane. (PDF)

## Author Contributions

**Conceptualization:** Andrew Thorne, Akanksha Bansal, Andrei Georgescu, Robert Newton.

**Formal analysis:** Robert Newton.

**Funding acquisition:** Robert Newton.

**Investigation:** Andrew Thorne, Akanksha Bansal, Amandah Necker-Brown, Mahmoud M. Mostafa, Alex Gao, Andrei Georgescu, Cora Kooi.

**Methodology:** Andrew Thorne, Robert Newton.

**Project administration:** Robert Newton.

**Resources:** Richard Leigh.

**Supervision:** Richard Leigh, Robert Newton.

**Visualization:** Andrew Thorne.

**Writing – original draft:** Andrew Thorne, Mahmoud M. Mostafa, Robert Newton.

**Writing – review & editing:** Andrew Thorne, Akanksha Bansal, Amandah Necker-Brown, Mahmoud M. Mostafa, Alex Gao, Andrei Georgescu, Cora Kooi, Richard Leigh, Robert Newton.

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
