## [Decision Letter · Decision Letter 0]

7 Feb 2023

PONE-D-22-30303Differential regulation of BIRC2 and BIRC3 expression by inflammatory cytokines and glucocorticoids in pulmonary epithelial cellsPLOS ONE

Dear Dr. Newton,

Thank you for submitting your manuscript to PLOS ONE. After careful consideration, we feel that it has merit but does not fully meet PLOS ONE’s publication criteria as it currently stands. Therefore, we invite you to submit a revised version of the manuscript that addresses the points raised during the review process.

I have assessed your submission, and I agree with the reviewer's comments about the text fluency of the manuscript. I, therefore, request that, besides addressing all the points raised by the reviewers, you revise the text to improve the overall readability of the text, providing fluency and avoiding repetition.

We look forward to receiving your revised manuscript.

Kind regards,

Aristóbolo M Silva

Academic Editor

PLOS ONE

Journal Requirements:

Reviewers' comments:

Reviewer's Responses to Questions

**Comments to the Author**

1. Is the manuscript technically sound, and do the data support the conclusions?

Reviewer #1: Yes

Reviewer #2: Yes

2. Has the statistical analysis been performed appropriately and rigorously? 

Reviewer #1: Yes

Reviewer #2: Yes

3. Have the authors made all data underlying the findings in their manuscript fully available?

Reviewer #1: No

Reviewer #2: Yes

4. Is the manuscript presented in an intelligible fashion and written in standard English?

Reviewer #1: No

Reviewer #2: Yes

5. Review Comments to the Author

Reviewer #1: This article aims to study the regulation of the expression of BIRC2 and BIRC3 in bronchial epithelial cell lines cultured in the presence of pro-inflammatory cytokines (IL-1β or TNFα), in the presence or the absence of the anti-inflammatory compounds dexamethasone or budesonide.

The results shown are interesting but the article is very difficult to read and follow. This could be improved by an introduction in result paragraphs specifying the objectives of the experiments carried out. Moreover, concluding sentence at the end of the result paragraphs could help in the understanding of the results.

I did not understand why the authors analyzed the influence of INFγ (Fig S2D). Since INFγ is no longer used in later experiments and these results are not used for discussion, I believe they are not necessary (line 209-214).

Figure legends: could author indicate the number of experiments (instead of “N”)

Why the authors analyzed the regulation of BIRC2 and BIRC3 expression in pHBEC grown as both submersion culture or as highly differentiated cells? Did they expect different mechanisms of regulation in function of the differentiation state? It seems to me that the results are quite similar, whatever the cell line: A549, pHBEC-SC or pHBEC-ALI and also BEAS-2B and Calu-3. It should be possible to simplify the description of the results in order to avoid repetition and provide fluency (results paragraph 1 and 2).

Figure 3D showed that the IL-1β and TNFα-mediated overexpression of BIRC2 mRNA is prevented by IκBαΔN expression whereas it is not affected by PS-1145 (Figure 4A) that blocks NF-κB. Could authors explain these results. Moreover, PS-1145, as well as ML-120B reduced BIRC2 protein expression in TNF-α-treated cells while PS-1145 did not modify the level of expression of the mRNA. These results have to be explained.

A concluding sentence concerning the role of NF-κB in BIRC3 and BIRC2 expression at the end of the result paragraph 4 could greatly improve the understanding of the results.

Figure 5D. I think that there are errors in the indications of the presence or not of siGR and siCTL.

In the result paragraph 6, the authors analyzed the expression of BIRC3 and BIRC2 proteins in IL-1β or TNFα-treated cells in the presence of absence of cycloheximide. The author should introduce this paragraph and indicate the objectives of these experiments. What is the question addressed? Why did the authors add CHX 1 hours after the cytokine? CHX treatment prevented the IL-1β or TNFα-mediated increase of BIRC3 expression and in a lower extend BIRC2 expression and even decreased BIRC2 protein expression in TNFα-stimulated cells. Again, the authors should give the conclusion of the experiments to help in the understanding of the article. Then, the authors added CHX 6 hours after the cytokine. I did not understand why they did this experiment? What is the question addressed? These points could be precised in the manuscript.

In the last result paragraph, the author analyzed the mechanisms of BIRC2 and BIRC3 degradation? Since the degradation was only observed in the condition of TNFα + CHX treatment (Figure 6B), why did they analyze the condition of IL1β + CHX treatment and why did they analyze the role of proteasome inhibitor on IL1β- or TNFα-induced expression of BIRC3 and BIRC2 ? It is well known that NF-κB activation required proteasome-mediated degradation of IκBα. Since IL1β- or TNFα-induced expression of BIRC3 and BIRC2 is dependent on NF-κB, it seems logical that proteasome inhibitors block the induction of BIRC3 and BIRC2 expression.

The results show that BIRC2 and BIRC3 proteins is degraded 6-7 hours after stimulation of cells with TNFα but not after IL-1β stimulation. This could argue in a favor of a role of BIRC2 and BIRC3 in TNFR signaling pathway but not in IL-1βR signaling pathway. Since BIRC2 is an important cell signaling intermediate in TNFR signaling pathway, we may hypotheze that the degradation of BIRC2 is a feed-back regulatory mechanism in order to regulate TNFR signaling pathway in case of sustain stimulation of receptor. BIRC2 in an E3-ubiquitine ligase able to self-ubiquitinate and stimulate its own degradation and also able to promote the ubiquitinatation and degradation of BIRC3 (in case of Smac mimetic exposure). This point could be discussed.

Reviewer #2: In this excellent study, Thorne et al., from Newton’s group sought to dissociate the redundant roles of BIRC2 and BIRC3 in the regulation of inflammatory signaling in the presence or absence of glucocorticoids (GC) in physiologically relevant cells, airway epithelial cells. The authors used both pulmonary epithelial cell lines (A549, BEAS2B, Calu-3 cells) and primary human bronchial epithelial cells (pHBECs cultured either in submersion culture or ALI). To this end, responses to cytokines (TNF, IL1, IFNg) and GC (Dexamethasone, Budesonide) have been examined in detail. The role of NF-kB in BIRCs expression induced by cytokines has been demonstrated using various strategies such as siRNA, DN, and IKK inhibitors. The involvement of the glucocorticoid receptor in BRCs expression induced by GC has been also demonstrated using GR inhibitor or siRNA GR. The involvement of protein degradation in TNF effects on BRCs was explored and demonstrated using CHX and MG compound. The authors clearly demonstrated the differential regulation of BIRC3 and BIRC2 in epithelial cells by inflammatory cytokines and GC where the constitutive expression of BIRC2 is consistent with acute early events and the inducible expression of BIRC3 is more consistent with later onset roles. The authors also demonstrated that BIRC3 expression is induced by GC and suggested that BIRC3 plays key protective roles in inflammatory signaling. Overall, the study is well-designed, comprehensive and the Ms is well-written. Some minor issues are presented below.

Minor comments:

- While the authors suggested a protective role of BIRC3, some studies showed a correlation between BRC3 and asthma severity (PMID: 27304223). Further discussion is needed.

- While most of the supra-additive effects of IL1b and GC were seen at 24 hr (Fig. 2A), most of the western blot studies (Figs 3C, 5B and FC) were conducted at 6 hr where not clear supra-additive effects were observed. Some justification is needed.

- Figures: Fig 5D, western blot: Experimental conditions under bands, 3, 6, and 9 are not clear. Are those triplicates? Conditions with IL1B only were not presented while stated in the Fig 5D legend.

- Introduction section: Second paragraph, the opposite/contradicting role of BIRCs on the regulation of NF-kB in canonical (line 54-64) versus non-canonical pathways (line 64-67 where the term BRC2/3 inhibition is not clear) is a little bit confusing.

- Due the high number of data and variables in this study, the addition of a summary table highlighting all the findings in one place will enhance the clarity of this study.

- Similarly, a graph depicting the conclusion of the authors should help reinforcing this critical concept and findings generated by this study.

6. PLOS authors have the option to publish the peer review history of their article (what does this mean?). If published, this will include your full peer review and any attached files.

Reviewer #1: No

Reviewer #2: No

---

## [Author Response · Author response to Decision Letter 0]

27 Mar 2023

Please see the attached file for our full response to the Editor and reviewers.

---

## [Decision Letter · Decision Letter 1]

12 Apr 2023

PONE-D-22-30303R1Differential regulation of BIRC2 and BIRC3 expression by inflammatory cytokines and glucocorticoids in pulmonary epithelial cellsPLOS ONE

Dear Dr. Newton,

Thank you for submitting your manuscript to PLOS ONE. One of our reviewers still has raised concerns regarding the article, specifically citing difficulties in following the manuscript and issues with readability.

The reviewer has pointed out that the current version of the manuscript is difficult to follow, making it challenging to read. Additionally, the reviewer has noted that some results are repeated in different figures, and has suggested that the article could be further simplified.

In general, the reviewer has recommended that several figures could be deleted to improve clarity and conciseness, and has raised concerns about contrasting results between Figs. 2A, B and Fig. 6.

Please review the comments from the reviewer and make the necessary changes as requested.

We look forward to receiving your revised manuscript.

Kind regards,

Aristóbolo M Silva

Academic Editor

PLOS ONE

Journal Requirements:

Reviewers' comments:

Reviewer's Responses to Questions

**Comments to the Author**

1. If the authors have adequately addressed your comments raised in a previous round of review and you feel that this manuscript is now acceptable for publication, you may indicate that here to bypass the “Comments to the Author” section, enter your conflict of interest statement in the “Confidential to Editor” section, and submit your "Accept" recommendation.

Reviewer #1: (No Response)

Reviewer #2: All comments have been addressed

2. Is the manuscript technically sound, and do the data support the conclusions?

Reviewer #1: Yes

Reviewer #2: Yes

3. Has the statistical analysis been performed appropriately and rigorously? 

Reviewer #1: Yes

Reviewer #2: Yes

4. Have the authors made all data underlying the findings in their manuscript fully available?

Reviewer #1: Yes

Reviewer #2: Yes

5. Is the manuscript presented in an intelligible fashion and written in standard English?

Reviewer #1: Yes

Reviewer #2: Yes

6. Review Comments to the Author

Reviewer #1: The authors answered to my concerns. However, the article is still difficult to follow. I agree that replication of the results adds robustness but it also brings heaviness and makes the article difficult to read.

1. Figure S1 and Figure 3B both demonstrated that IL-1β and TNFα both induced the activation of NF-κB in the concentrations used in the study. I think that the supplementary Figure 1 can be removed. Then, the authors analyzed the contribution of NF-κB in IL-1β and IL-1β + DEX-induced BIR2 and BIRC3 mRNA expression by using RELA-directed siRNA or IκBα super-repressor. They showed that IL-1β and IL-1β + DEX-induced increased of BIRC2 and BIRC3 mRNA are, at least partly, NF-κB dependent. Then, they confirmed that the increase in BIRC3 protein expression also depended on NF-κB (Figure 3 and S6). The same results were observed in Figure 3C and S6. Could authors simplify the text by describing the two Figure in the same sentence. Since IL1B and IL1B + DEX did not enhanced BIRC2 protein expression (as demonstrated in Figure 2), I think that the authors could simplify the Figure 3 and remove the analysis of BIRC2 protein expression.

2. In Figure 4A, the authors confirmed that NF-κB is involved in the induction of BIRC3 mRNA and protein expression in IL1B, TNFα, IL1B + DEX and TNF + DEX-treated cells by using the IKK2 inhibitor PS-1145. By contrast, PS-1145 did not decrease the induction of BIRC2 mRNA in response to the stimuli. The analysis of BIRC3 protein expression confirmed the involvement of NF-κB. Again, since IL1B and IL1B + DEX did not enhanced BIRC2 protein expression (as demonstrated in Figure 2), I think that the authors could simplify the figure and remove the analysis of BIRC2 protein expression. Since the conclusions of the result paragraph 3 and 4 (Figure 3 and 4) are the same, e.i. the induction of BIRC3 mRNA and protein expression in IL1B, TNF, IL1B + DEX and TNF + DEX treated cells depended on NF-kB, the two paragraphs could be combined.

Line 342: “Dexamethasone-induced BIRC3 mRNA expression was not changed by PS-1145”. I do not agree with this sentence. Although statistically non-significant, the modest induction of BIRC3 in response to dexamethasone (in the left and right panels) in decreased by PS-1145 (Figure 4).

3. The authors analyzed the stability of BIRC2 and BIRC3 proteins in TNFα and IL-1β treated cells (Figure 6). Cells were first treated with the agents for 1 hours, then incubated in the presence or absence of CHX. As expected, IL1b or TNF both enhanced BIRC3 protein expression. While 6 hrs of treatment with IL1B or TNFα did not modify the BIRC2 protein expression in Figure 2A and B, it seems increased BIRC2 protein expression in Figure 6B (quantitative analysis, lower panel)? Is that significant? Could authors explain these results?

4. Figure 8C, left panel reproduced the results of Figure 6B. The authors could simplify and remove this panel. Since the authors demonstrated that IL-1β did not destabilize BIRC2 and BIRC3 protein in Figure 6B, the analysis the importance of proteasome inhibitor on destabilization of BIRC proteins in IL-1β treated cells is not relevant and this can be removed.

Reviewer #2: The authors have adequately addressed all the comments. No further comments.

The authors have adequately addressed all the comments. No further comments.

7. PLOS authors have the option to publish the peer review history of their article (what does this mean?). If published, this will include your full peer review and any attached files.

Reviewer #1: No

Reviewer #2: No

---

## [Author Response · Author response to Decision Letter 1]

29 Apr 2023

Response to reviewer comments

Reviewer #1: The authors answered to my concerns. However, the article is still difficult to follow. I agree that replication of the results adds robustness but it also brings heaviness and makes the article difficult to read. 

Author response: Thanks for these remarks. In addition to specifically addressing the concerns raised below, we have also made various minor modifications to the text of the manuscript with the aim of further improving readability. 

1. Figure S1 and Figure 3B both demonstrated that IL-1β and TNFα both induced the activation of NF-κB in the concentrations used in the study. I think that the supplementary Figure 1 can be removed. 

Author Response: While both Fig 3B and Fig S1 do indeed show activation of the NF-κB reporter, the data shown in Fig S1 is necessary to first establish the maximal effective concentrations of IL1B and TNF that are then used in all subsequent experiments. This point was not clearly made in the prior version of the manuscript. Sorry! We have now reworked this part to make the importance of these data more readily apparent to the reader. Thus, Fig S1 represents a necessary piece of initial data and provides the necessary rationale for using 1 ng/ml of IL1B and 10 ng/ml of TNF. We therefore prefer to retain this figure as Supplemental Fig S1.

Then, the authors analyzed the contribution of NF-κB in IL-1β and IL-1β + DEX-induced BIR2 and BIRC3 mRNA expression by using RELA-directed siRNA or IκBα super-repressor. They showed that IL-1β and IL-1β + DEX-induced increased of BIRC2 and BIRC3 mRNA are, at least partly, NF-κB dependent. Then, they confirmed that the increase in BIRC3 protein expression also depended on NF-κB (Figure 3 and S6). The same results were observed in Figure 3C and S6. Could authors simplify the text by describing the two Figure in the same sentence. Since IL1B and IL1B + DEX did not enhanced BIRC2 protein expression (as demonstrated in Figure 2), I think that the authors could simplify the Figure 3 and remove the analysis of BIRC2 protein expression.

Author Response: We have condensed this part of the text as requested by the reviewer. We have also removed Figure 3C bottom panel (analysis of BIRC2 protein expression) and have also condensed the relevant parts of the text relating to these data.

2. In Figure 4A, the authors confirmed that NF-κB is involved in the induction of BIRC3 mRNA and protein expression in IL1B, TNFα, IL1B + DEX and TNF + DEX-treated cells by using the IKK2 inhibitor PS-1145. By contrast, PS-1145 did not decrease the induction of BIRC2 mRNA in response to the stimuli. The analysis of BIRC3 protein expression confirmed the involvement of NF-κB. Again, since IL1B and IL1B + DEX did not enhanced BIRC2 protein expression (as demonstrated in Figure 2), I think that the authors could simplify the figure and remove the analysis of BIRC2 protein expression. 

Author Response: As requested, we have deleted the protein expression analysis for BIRC2 when cells were treated with PS-1145. The relevant parts of the text have also been amended to reflect this change. 

Since the conclusions of the result paragraph 3 and 4 (Figure 3 and 4) are the same, e.i. the induction of BIRC3 mRNA and protein expression in IL1B, TNF, IL1B + DEX and TNF + DEX treated cells depended on NF-kB, the two paragraphs could be combined.

Author Response: As requested, we have condensed the text to include the effect of NF-κB on BIRC expression as a single section of results.

Line 342: “Dexamethasone-induced BIRC3 mRNA expression was not changed by PS-1145”. I do not agree with this sentence. Although statistically non-significant, the modest induction of BIRC3 in response to dexamethasone (in the left and right panels) in decreased by PS-1145 (Figure 4).

Author Response: We thank the reviewer for this keen observation. We agree that there may be a genuine effect where cells treated with Dex + PS-1145 as this does appear to result in a lower level of BIRC3 mRNA. However, we note that this effect was not significant. Nevertheless, inhibition of NF-κB may well reduce the basal expression of BIRC3 as supporting evidence can be seen in Figs 3C, 3D and 4A where there are also modest reductions in BIRC3 expression following NF-κB is inhibition in the unstimulated groups. We have revised the sentence that was flagged by the reviewer and have commented on the possible role of basal NF-κB activity, i.e. even in unstimulated cells. 

3. The authors analyzed the stability of BIRC2 and BIRC3 proteins in TNFα and IL-1β treated cells (Figure 6). Cells were first treated with the agents for 1 hours, then incubated in the presence or absence of CHX. As expected, IL1b or TNF both enhanced BIRC3 protein expression. While 6 hrs of treatment with IL1B or TNFα did not modify the BIRC2 protein expression in Figure 2A and B, it seems increased BIRC2 protein expression in Figure 6B (quantitative analysis, lower panel)? Is that significant? Could authors explain these results?

Author Response: The modest effect on BIRC2 expression shown in Figure 6 that is mentioned by the referee was not significant. As noted by the reviewer, there is some variability when documenting BIRC2 expression throughout the manuscript. We believe that this is primarily due to the inherent variability in western blotting combined with small biological effects and relatively small samples sizes. It is worth noting that any changes in BIRC2 protein expression were less than 2-fold. Western blotting, as semi-quantitative, or comparative, technique (& depending on operator skill) is not really robust enough to clearly document such small expression changes. While some blots appear to support a slight increase in BIRC2, others don’t. Overall, any changes were difficult to consistently capture and this resulted in non-significant outcomes. 

4. Figure 8C, left panel reproduced the results of Figure 6B. The authors could simplify and remove this panel. Since the authors demonstrated that IL-1β did not destabilize BIRC2 and BIRC3 protein in Figure 6B, the analysis the importance of proteasome inhibitor on destabilization of BIRC proteins in IL-1β treated cells is not relevant and this can be removed.

Author Response: With the aim of creating a concise manuscript, we have removed Figure 8C left- and middle-panel. As a consequence, several lines of text have been removed from this section.

Reviewer #2: The authors have adequately addressed all the comments. No further comments.

The authors have adequately addressed all the comments. No further comments.

Author Response: We thank the reviewer for this.

---

## [Decision Letter · Decision Letter 2]

24 May 2023

Differential regulation of BIRC2 and BIRC3 expression by inflammatory cytokines and glucocorticoids in pulmonary epithelial cells

PONE-D-22-30303R2

Dear Dr. Newton,

We’re pleased to inform you that your manuscript has been judged scientifically suitable for publication and will be formally accepted for publication once it meets all outstanding technical requirements.

Kind regards,

Aristóbolo M Silva

Academic Editor

PLOS ONE

Additional Editor Comments (optional):

Reviewers' comments:

Reviewer's Responses to Questions

**Comments to the Author**

1. If the authors have adequately addressed your comments raised in a previous round of review and you feel that this manuscript is now acceptable for publication, you may indicate that here to bypass the “Comments to the Author” section, enter your conflict of interest statement in the “Confidential to Editor” section, and submit your "Accept" recommendation.

Reviewer #1: All comments have been addressed

2. Is the manuscript technically sound, and do the data support the conclusions?

Reviewer #1: Yes

3. Has the statistical analysis been performed appropriately and rigorously? 

Reviewer #1: Yes

4. Have the authors made all data underlying the findings in their manuscript fully available?

Reviewer #1: Yes

5. Is the manuscript presented in an intelligible fashion and written in standard English?

Reviewer #1: Yes

6. Review Comments to the Author

Reviewer #1: (No Response)

7. PLOS authors have the option to publish the peer review history of their article (what does this mean?). If published, this will include your full peer review and any attached files.

Reviewer #1: No

---

## [Editor Report · Acceptance letter]

26 May 2023

PONE-D-22-30303R2 

Differential regulation of BIRC2 and BIRC3 expression by inflammatory cytokines and glucocorticoids in pulmonary epithelial cells 

Dear Dr. Newton:

I'm pleased to inform you that your manuscript has been deemed suitable for publication in PLOS ONE. Congratulations! Your manuscript is now with our production department. 

Kind regards, 

on behalf of

Dr. Aristóbolo M Silva 

Academic Editor

PLOS ONE